# STRUCTURED ENERGY NETWORK
# AS A DYNAMIC LOSS FUNCTION
## A CASE STUDY WITH MULTI-LABEL CLASSIFICATION

## ABSTRACT

We propose `SEAL` which utilizes an energy network as a trainable loss function for a simple feedfoward network. Structured prediction energy networks (SPENs) (Belanger & McCallum, 2016; Gygli et al., 2017) have shown that a neural network (i.e. energy network) can learn a reasonable energy function over the candidate structured outputs. We find that rather than using SPENs as a prediction network, using it as a trainable loss function is not only computationally more efficient but also better performing. As the energy function is trainable, we propose `SEAL` to be dynamic so it can adapt the energy function to focus on the region where feed-forward models will be affected the most at the time point of model update. We find this to be effective in an ablation study comparing `SEAL` to the static version (§5) where the energy function is fixed after pretraining. We show the relation to previous work on the joint optimization model of energy network and feedforward model (INFNET) as we show that it is equivalent to `SEAL` using margin-based loss if INFNET relaxes their loss function. Based on the unique architecture of `SEAL`, we further propose a variant of `SEAL` that utilizes noise contrastive ranking (NCE) loss that by itself does not perform well as a structured energy network, but embodied in `SEAL`, it shows the greatest performance among the variants we study. We demonstrate the effectiveness of `SEAL` on 7 feature-based and 3 text-based multi-label classification datasets. The best version of `SEAL` that uses NCE ranking method (`SEAL-NCE`) achieves close to 2.85, 2.23 respective F1 point gain on average over cross-entropy and INFNET for feature-based datasets, excluding one dataset that has an excessively large gain. Lastly, examining whether the proposed framework is effective on a large pre-trained model as well, we observe `SEAL-NCE` achieving 0.87, 0.68 respective F1 point gain in average over cross-entropy and INFNET with BERT-based adapter model on text datasets.

## 1 INTRODUCTION

Structured prediction is a popular machine learning task wherein the model learns a mapping function from an input $\mathbf{x}$ to a multivariate-structured output $\mathbf{y}$. Popular examples of this include image segmentation (Müller, 2014), extracting parse trees or semantic role labels from a text (Palmer et al., 2010), and multi-label classification (Belanger & McCallum, 2016; Gygli et al., 2017). In structured prediction, the output space $\mathcal{Y}$ is often extremely large. For example, in multi-label classification, the size of $\mathcal{Y}$ is $2^L$ where model needs to predict output $\mathbf{y} \in \{0, 1\}^L$. There are two key aspects that a model for structured prediction needs to balance: statistical efficiency and computational efficiency. Most models for structured prediction can be categorized into two categories: the feed-forward approach, wherein one learns a neural network that models the direct mapping between the input and the structured output, and the structured approach, in which the model explicitly models the interactions (structure) in the output space.

The feed-forward approach learns all dimensions of $\mathbf{y}$ jointly in a conditionally independent manner given input $\mathbf{x}$, relying on the representational power of the network to capture dependencies in output structure. While the feed-forward approach is computationally efficient, since it does not capture relationships in the label space, it lacks statistical efficiency. The traditional structured approaches, on the other hand, model a joint probability distribution $P(\mathbf{x}, \mathbf{y})$ that can capture the label relationships. However, due to the intractability of modeling the full joint distribution which

captures every possible interaction of output space, these approaches resort to limiting the interaction terms to local subsets of the output space (Lafferty et al., 2001; Ghamrawi & McCallum, 2005). In order to make inference for structured models more efficient, a recent line of work replaces the joint probability with a structured energy function $E(\mathbf{x}, \mathbf{y})$ (LeCun et al., 2006), which can be thought of as an unnormalized probability distribution, allowing the model to learn arbitrary global dependencies in the output space. There have been several works that propose efficient approximate inference procedures for structured energy networks using gradient based inference (Belanger et al., 2017; Gygli et al., 2017; Rooshenas et al., 2019), hence also called 'prediction networks'. These models showed noticeable gains in predictive performance over feed-forward models and graphical models that assume partial structure such as full pairwise potential (Chen et al., 2015; Schwing & Urtasun, 2015). Despite these efforts, the inference for energy-based models still remains relatively inefficient when compared to the feed-forward approach (Tu & Gimpel, 2019). Moreover, in our experience, the models using gradient based inference (GBI) are finicky to train as the training also utilizes GBI, requiring numerous hyperparameters: step size, number of iterations for GBI, and initial point to begin GBI. This raises a question: *Can the energy network be used in a way that is as expressive as a full joint probability, as efficient at inference as a feed-forward approach, and also stable and easy to train?* We believe that using structured energy networks as a parameterized dynamic loss function for feed-forward networks, instead of a prediction network, can fulfill all these requirements.

In this paper, we propose the **S**tructured **E**nergy **A**s **L**oss (SEAL) framework that uses a trainable structured energy network (SEN)[1] as a loss function guiding the training of a feed-forward network. The key idea is to provide the feed-forward network access to rich relationships in the output space through a learned loss function. We also propose to learn SEN in a dynamic fashion by adjusting the energy function to be confident with the most up-to-date outputs of the feed-forward network. We show that learning the loss function dynamically leads to more efficient, more stable, and better performance. SEAL can be viewed as general-purpose framework where one can plugin various loss functions and architectures to train SEN as well as the feed-forward network. Through experiments (§4), we analyze the effect of applying different energy losses (e.g. margin-based, regression-based) within SEAL framework. We also propose the noise-contrastive ranking loss (NCEranking) for SEN which performs the best within the SEAL framework.

To summarize, we introduce a general framework SEAL that interprets structured energy networks (SEN) as a dynamic loss functions. Through empirical evaluation on the task of multi-label classification, we analyse the impact of various loss functions for updating SEN. Finally, we propose an NCE ranking loss that is uniquely suited for the SEAL framework, and demonstrate its superior performance on 7 feature-based as well as 3 text-based multi-label classification datasets, when compared to simple feed-forward approach and various energy based models.

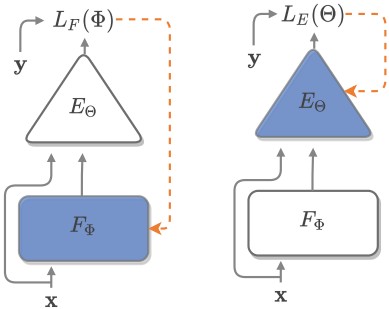

Figure 1: Overview of the SEAL framework. The figure on the right shows the update to the energy network (equation 4) and the figure on the left shows the update to the feed-forward network (equation 3).

Algorithm 1: SEAL Algorithm

**Require:** $(\mathbf{x}, \mathbf{y})$: Training Instance
**Require:** $F_\Phi$: Feedforward Network
**Require:** sampling: True/False flag
**Require:** optimizer$_\Theta$, optimizer$_\Phi$
**Require:** $T$: No. of steps
  $t \leftarrow 0$
  $\Theta_0, \Phi_0 \leftarrow$ Random initialization
  **while** $t < T$ **do**
    **if** sampling **then**
      $S \leftarrow \{\mathbf{y}^{(i)}, i = 1, \ldots, K | \mathbf{y}^{(i)} \sim F_{\Phi_t}(\mathbf{x})\}$
    **else**
      $S \leftarrow \{F_{\Phi_t}(\mathbf{x})\}$            ▷ singleton set
    $t \leftarrow t + 1$
    Update $\Theta$ as Eqn. 3 using $\tilde{\mathbf{y}} \in S$ and optimizer$_\Theta$
    Update $\Phi$ as Eqn. 4 using optimizer$_\Phi$

---

[1]The term comes from structured prediction energy network, minus 'prediction', as we do not use it for prediction anymore. For brevity, we use terms 'energy network' and 'strcutured energy newtork' interchangeably.

## 2  STRUCTURED ENERGY NETWORK AS LOSS (SEAL)

This section first describes the proposed SEAL framework. Let $\mathcal{X}$ denote the input space, $\mathcal{Y} = \{0, 1\}^L$ the output space, and $\tilde{\mathcal{Y}} = [0, 1]^L$ the continuous relaxation of $\mathcal{Y}$. Then the structured energy $E_\Theta : \mathcal{X} \times \tilde{\mathcal{Y}} \to \mathbb{R}$ is defined to be a parameterized function of the input and the continuous relaxation of the output. The feedforward network $F_\Phi : \mathcal{X} \to \tilde{\mathcal{Y}}$ is defined to be a neural network that maps an input to the continuous relaxation of the output space. We denote $j$-th training instance pair as $(\mathbf{x}^{(j)}, \mathbf{y}^{(j)}) \in \mathcal{D}$ and $y_i \in \{0, 1\}$, $\tilde{y}_i \in [0, 1]$ to denote $i$-th label dimension where $i \in \{1, \ldots, L\}$.

As shown in Figure 1, SEAL consists two loss functions: the energy loss $L_E$ that trains structured energy network $(\Theta)$, and feedforward loss $L_F$ that guides the training of the feedforward network $(\Phi)$. We first discuss how structured energy network implicitly affects training of the feedforward network by defining $L_F(\Phi)$. Given a training instance $(\mathbf{x}, \mathbf{y})$, the feedforward loss is defined as:

$$L_F(\Phi) = \lambda_1 E_\Theta(\mathbf{x}, F_\Phi(\mathbf{x})) - \lambda_2 \sum_{j=1}^{L} \left[ y_i \log F_\Phi(\mathbf{x})_i + (1 - y_i) \log \left(1 - F_\Phi(\mathbf{x})_i\right) \right] . \qquad (1)$$

Here, the first term involving the energy captures interaction across label space whereas binary cross-entropy considers the predictions of each label independently (More details in Appendix B).

Since the training of feedforward network depends on the quality of the parameterized energy network, it is critical to find the parameters $\Theta$ that produce the best loss surface for training the feedforward network. This can be done in two ways: by first training the energy separately, or by training the energy and the feedforward network simultaneously. SEAL uses the latter, which is shown to perform better (see §5) than the former. We denote the former method as SEAL-static for distinction .

In SEAL-static, we first estimate the energy network parameter $\widehat{\Theta}$ over training data and optimize $\Phi$ by plugging fixed $\widehat{\Theta}$ into $L_F$. Given $(\mathbf{x}, \mathbf{y}) \in \mathcal{D}$, $\Phi$ is trained as equation 2.

$$\min_{\Phi} \frac{1}{|\mathcal{D}|} \sum_{\mathcal{D}} L_F(\Phi) \quad \text{s.t.} \quad \widehat{\Theta} = \arg\min_{\Theta} \frac{1}{|\mathcal{D}|} \sum_{\mathcal{D}} L_E(\mathbf{x}, \mathbf{y}, \tilde{\mathbf{y}}; \Theta) \qquad (2)$$

We finally present SEAL which learns $\widehat{\Theta}_t$ dynamically at step $t$. To do so, we alternate the optimization steps of $\Theta$ and $\Phi$. Given a training instance $(\mathbf{x}, \mathbf{y}) \in B_t$, a complete training step is given as:

$$\Theta_t \quad \leftarrow \quad \Theta_{t-1} - \nabla_\Theta \frac{1}{|B_t|} \sum_{B_t} L_E \left(\mathbf{x}, \mathbf{y}, F_{\Phi_{t-1}}(\mathbf{x}); \Theta\right) \qquad (3)$$

$$\Phi_t \quad \leftarrow \quad \Phi_{t-1} - \nabla_\Phi \frac{1}{|B_t|} \sum_{B_t} L_F(\Phi) \qquad (4)$$

Note that, from (2) to (3), arbitrary $\tilde{\mathbf{y}}$ got replaced with $F_{\Phi_{t-1}}(\mathbf{x})$ in estimating $\Theta_t$ so that estimation of energy surface depends on model output $F_{\Phi_{t-1}}(\mathbf{x})$. Further, the update step for feedforward network $\Phi$ relies on $\Theta_t$ rather than static $\widehat{\Theta}$. In both SEAL and SEAL-static, test-time predictions are performed solely using the feedfoward network as $\hat{y}_i = \mathbf{1}(F_\Phi(x)_i \geq 0.5)$.

The motivation for proposing dynamic loss function in SEAL is as following. It would be useful to learn perfect energy surface, if possible, by estimating $\widehat{\Theta}$ that is static. However, doing so is challenging as we are estimating energy surface on the joint space of $\mathbf{x}, \tilde{\mathbf{y}}$, a high-dimensional continuous space with limited data and resource. Instead, we hypothesize that it is more important to concentrate resource in depicting accurate energy surface around the current input, output pair $(\mathbf{x}, F_\Phi(x))$. In SEAL, utilizing the fact that we have access to trainable loss function, we dynamically adapt energy surface so that the region of interest for training feedforward network is well represented.

We now discuss various energy loss (margin-based, regression-based, and contrastive-sampling-based) that can be plugged into $L_E$ in SEAL. For brevity, we use $L_E(\Theta)$ and $L_E(\mathbf{x}, \mathbf{y}, \tilde{\mathbf{y}}; \Theta)$ interchangeably where again $(\mathbf{x}, \mathbf{y})$ denotes labeled data and $\tilde{\mathbf{y}}$ denotes probability vector. We defer the description of specific energy network structure to the experiment section (§4) as SEAL framework can work with arbitrary network structures.

**Margin-based ($\mathbf{L}_{E-\mathbf{margin}}$)** SPEN (Belanger & McCallum, 2016) learned structured energy network with SSVM loss (Taskar et al., 2004; Tsochantaridis et al., 2004) so that energy functions learn to have sufficient energy difference, larger than the margin $\Delta(\tilde{\mathbf{y}}, \mathbf{y})$, between arbitrary output $\tilde{\mathbf{y}}$ and true output $\mathbf{y}$. To examine the effect of the margin-based loss in SEAL, we follow SPEN and utilize the SSVM loss as

$$\mathrm{L}_{E-\mathrm{margin}} = \sum_{\mathbf{x},\mathbf{y}} \max_{\tilde{\mathbf{y}}} \left[ \Delta(\tilde{\mathbf{y}}, \mathbf{y}) - E_\Theta(\mathbf{x}, \tilde{\mathbf{y}}) + E_\Theta(\mathbf{x}, \mathbf{y}) \right]_+ \tag{5}$$

**Regression-based ($\mathbf{L}_{E-\mathbf{regression}}$)** Deep Value Network (DVN) (Gygli et al., 2017) attempts to learn an energy network which directly outputs a score that is similar to the metric $s(\tilde{\mathbf{y}}, \mathbf{y})$ of interest, such as F1 score, that compares arbitrary $\tilde{\mathbf{y}}$ with true output $\mathbf{y}$. Following DVN, by making score $s(\cdot)^2$ and $-E(\cdot)$ to be between $[0, 1]$, we express regression loss as cross entropy

$$\mathrm{L}_{E-\mathrm{regression}} = -s(\tilde{\mathbf{y}}, \mathbf{y}) \log - E_\Theta(\mathbf{x}, \tilde{\mathbf{y}}) - (1 - s(\tilde{\mathbf{y}}, \mathbf{y})) \log(1 + E_\Theta(\mathbf{x}, \tilde{\mathbf{y}})). \tag{6}$$

**Noise-contrastive ranking ($\mathbf{L}_{E-\mathbf{NCEranking}}$ and $\mathbf{L}_{E-\mathbf{ranking}}$)** In SEAL, we are interested in capturing the output region that feedforward network has a high probability. We ask, rather than just taking a single point $\tilde{\mathbf{y}} = F_\Phi(\mathbf{x})$ from a feedforward network, whether sampling many discrete binary vectors from $\tilde{\mathbf{y}}$ and taking all those samples into consideration for learning $E_\Theta$ could lead in better estimation of energy surface. Motivated from noise contrastive estimation (NCE) (Ma & Collins, 2018), we ask whether energy network $E_\Theta$ trained to contrast the true output $\mathbf{y}$ from groups of $K$ negative samples drawn from feedforward output $F_\Phi(\mathbf{x})$ can induce a good loss for teaching $\Phi$. The intuition is as the $\tilde{\mathbf{y}} = F_\Phi(\mathbf{x})$ becomes better and better, the loss surface teaching feedforward network should become more fine grained as $\tilde{\mathbf{y}}$ and samples from $\tilde{\mathbf{y}}$ will be already close to the true $\mathbf{y}$.

Before we discuss $\mathrm{L}_{E-\mathrm{NCEranking}}$ of our choice, we first review original form of NCE ranking loss from Ma & Collins (2018). For $K$ samples $\mathbf{y}^{(k)} \sim P_N$, $k = 1, \ldots, K$ drawn from noise distribution $P_N$ and rewriting $\mathbf{y}$ as $\mathbf{y}^{(0)}$ without loss of generality, NCE ranking loss is defined as

$$\log \frac{\exp\left(s(\mathbf{x}, \mathbf{y}^{(0)}; \Theta)\right)}{\sum_{k=1}^{K} \exp\left(s(\mathbf{x}, \mathbf{y}^{(k)}; \Theta)\right)}, \quad \text{where} \quad s(\mathbf{x}, \mathbf{y}; \Theta) = -E_\Theta(\mathbf{x}, \mathbf{y}) - \log P_N(\mathbf{y}). \tag{7}$$

Minimization of this loss makes $P(y|x; \Theta) = \frac{\exp(-E_\Theta(x,y))}{\sum_{y \in \mathcal{Y}} \exp(-E_\Theta(x,y))}$ as an unbiased estimator of true distribution $P(Y|X)$. Thus, given perfect energy $E(\cdot)$, minimizing $-E_\Theta(\mathbf{x}, \mathbf{y})$ with respect to $\mathbf{y}$ in $\mathrm{L}_F$ would be equivalent to maximizing the estimate of $-\log P(\mathbf{y}|\mathbf{x})$.

From the presented NCE ranking method, we propose to use $P(\mathbf{y}|\mathbf{x}; \Phi_t) = \prod_i P(\mathbf{y}_i|\mathbf{x}; \Phi_t)$ in place of $P_N(\mathbf{y})$. The novelty of this proposal is that we view the output of feedforward network that we train as a noise distribution, and that the energy model that contrasts noise distribution teaches the noise distribution as well through our SEAL framework (equation 3). In short, we get $\mathrm{L}_{E-\mathrm{NCEranking}}$ when we plug in $s(\mathbf{x}, \mathbf{y}; \Theta) = -E_\Theta(\mathbf{x}, \mathbf{y}) - \log P_\Phi(\mathbf{y}|\mathbf{x}; \Phi_t)$ into equation 7 , i.e .

$$\mathrm{L}_{E-\mathrm{NCEranking}} = \log \frac{\exp\left(s(\mathbf{x}, \mathbf{y}^{(0)}; \Theta)\right)}{\sum_{k=1}^{K} \exp\left(s(\mathbf{x}, \mathbf{y}^{(k)}; \Theta)\right)}, s(\mathbf{x}, \mathbf{y}; \Theta) = -E_\Theta(\mathbf{x}, \mathbf{y}) - \log P_\Phi(\mathbf{y}|\mathbf{x}; \Phi_t). \tag{8}$$

This achieves two benefits that we have in mind in using NCE. First benefit is that it captures the region of interest by sampling from the output distribution of $\Phi_t$. Second benefit is that it brings more efficient NCE method. NCE method is known to work best when the noise distribution is close to the data distribution (Gutmann & Hirayama, 2012) but not exactly the same. In our SEAL learning procedure, we hypothesize that $P(\mathbf{y}|\mathbf{x}; \Phi_t)$ becomes closer to the data distribution as training steps proceed.

Lastly, motivated that NCE can estimate the true distribution, we propose yet another loss function that can estimate the difference between the true probability $P(\mathbf{y}|\mathbf{x})$ and feedforward network probability. We show in appendix A, if we use plain ranking loss that sets $s(\mathbf{x}, \mathbf{y}; \Theta) = -E_\Theta(\mathbf{x}, \mathbf{y})$ in equation 7, unbiased estimator of $P(\mathbf{y}|\mathbf{x})$ now becomes $\frac{P_N(\mathbf{y}) \exp(-E_\Theta(\mathbf{x}, \mathbf{y}))}{\sum_{\mathbf{y} \in \mathcal{Y}} P_N(\mathbf{y}) \exp(-E_\Theta(\mathbf{x}, \mathbf{y}))}$ (9). This

---

[2]In this paper, we adopt soft F1 score $s(\tilde{\mathbf{y}}, \mathbf{y})$ from Gygli et al. (2017) that is defined on the continuous $\tilde{\mathbf{y}} \in [0, 1]^L$.

means if we take a derivative of $E_\Theta$ with respect to $\mathbf{y}$, then we are taking derivative with respect to $-\log P(\mathbf{y}|\mathbf{x}) + \log P_N(\mathbf{y})$. With the trick of replacing the $P_N(\mathbf{y})$ with $P(\mathbf{y}|\mathbf{x}; \Phi_t)$ in SEAL, we hypothesize that loss captured by energy network can focus only on the difference between true and feedforward probability which may end up being more informative surface. In short, we get plain ranking loss $\mathrm{L}_{E-\mathrm{ranking}}$ by plugging $s(\mathbf{x}, \mathbf{y}; \Theta) = -E_\Theta(\mathbf{x}, \mathbf{y})$ into equation 7 , resulting in

$$\mathrm{L}_{E-\mathrm{ranking}} = \log \frac{\exp\left(-E(\mathbf{x}, \mathbf{y}^{(0)}; \Theta)\right)}{\sum_{k=1}^{K} \exp\left(-E(\mathbf{x}, \mathbf{y}^{(k)}; \Theta)\right)}. \tag{10}$$

## 3 RELATED WORK

**Learning with dynamic loss:**  Wu et al. (2018) and Huang et al. (2019) attempt to learn a dynamic loss defined by neural network that is tailored to the task-specific metric $m$, such as BLEU or 0-1 accuracy. Both approaches try to to learn a loss function that can directly increase model's metric score on the validation set. Since $m$ is usually non-differentiable, Huang et al. (2019) view this problem as a reinforcement learning problem and attempts to learn a loss function that maximizes the expected reward: higher $m$ score in validation set. On the other hand, Wu et al. (2018) relax the $m$ to be differentiable and view the feed-forward model update and loss-model update as a connected chain of computation graph, so that loss function parameter ($\Theta$) gets updated in relation to feed-forward model ($\Phi$) improving the relaxed metric score.

There are three major differences between SEAL and the previous work. First, energy surface of SEAL is not limited to a specific parametric form, $\sigma(\mathbf{y}^T \Theta \log F_\Phi(x))$[3], as previous papers do. Instead, SEAL simply utilizes a neural network to express the loss function as $-E_\Theta(\cdot)$ which allows it to represent a much larger function class. Second, previous works require manual design of state vector. For example, the state vector used by Wu et al. (2018) consists of iteration number, validation accuracy, and current loss-function parameter $\Theta$. In contrast, SEAL only requires the input and output pair $(\mathbf{x}, F_\Phi(x))$ of the feed-forward network. Lastly, since these models were designed for multi-class classification, it is non-trivial to extend them to the multi-label classification or to the general structured prediction task wherein large number of labels interact.

**Structured Prediction Energy Networks:**  Structured prediction energy networks (Belanger & McCallum, 2016) and its variants (Gygli et al., 2017; Rooshenas et al., 2019) (referred to as SPENs from here on) learn an energy network $E_\Theta : \mathcal{X} \times \tilde{\mathcal{Y}} \to \mathbb{R}$ and predict the output using gradient-based inference (GBI) with the objective $\mathbf{y} = \arg\min_y E(\mathbf{x}, \mathbf{y})$. To train energy network, SPENs apply GBI adversarially in order to find $\mathbf{x}, \mathbf{y}$ pair that violates the minimality of the energy the most. We found this training procedure to be a bit unstable as its success depends on the initial point of GBI. We discuss the loss function for SPEN (Belanger & McCallum, 2016) and DVN (Gygli et al., 2017) in §2 and use these models as baselines in this work.

**Jointly learning $\Theta$ and $\Phi$:**  The Inference network (INFNET) proposed in Tu & Gimpel (2018) is the first work that tried to learn structured energy network and feedforward network jointly. As the name suggests, INFNET proposes a model that mimics the behaviour of SPEN by replacing the expensive gradient-based inference with feedforward networks. However, just like SPENs, INFNET also resorts to finding adversarial point that has the maximum energy violation, resulting in an adversarial framework similar to GAN (Goodfellow et al., 2014), where INFNET maximizes, and energy network minimizes the same margin based loss given in equation 5. INFNET has to serve as the adversarial sampler for training energy network. While the end goal of INFNET is an efficient inference for finding output with lowest energy, having to serve two roles, INFNET suffers from train-, test-time objective mismatch and Tu et al. (2020) resolves this by augmenting INFNET structure; Two separate feed-forward networks are used, one for prediction and the other as the adversarial sampler. However, INFNET's reliance on margin-based losses (marign-rescaled higne and perceptron) still leaves the approach to be adversarial, i.e. $\min_\Theta \max_\Phi \mathrm{L}_E(\mathbf{x}, \mathbf{y}, F_\Phi(x); \Theta)$.

In contrast, SEAL has completely independent loss functions $\mathrm{L}_E$ and $\mathrm{L}_F$ for updating the energy and feed-forward network. On feed-forward ($\Phi$) update, SEAL simply update the model in the direction

---

[3]The papers (Wu et al., 2018; Huang et al., 2019) use $\Phi$ instead of $\Theta$. We change it to $\Theta$ to put notations in consistent manner with this paper.

| Dataset | Domain | #Instances | | | #Labels | Input Type | Label Taxonomy |
|---------|--------|-------|-----|------|---------|------------|----------------|
| | | Train | Val | Test | | | |
| Expr FUN | Gene Ontology | 1636 | 849 | 1288 | 500 | Continuous | Forest |
| Spo FUN | Gene Ontology | 1600 | 837 | 1266 | 500 | Continuous | Forest |
| Bibtex | Text | 4407 | 1491 | 1497 | 159 | Binary | - |
| Cal500 | Music | 283 | 105 | 114 | 174 | Continuous | - |
| Delicious | Text | 9690 | 3207 | 3194 | 983 | Binary | - |
| Genbase | Biology | 398 | 132 | 132 | 27 | Binary | - |
| Eurlex-ev | Text | 11557 | 3876 | 3881 | 3993 | Binary | - |
| BGC | Text | 58715 | 14785 | 18394 | 142 | Raw Text | Forest |
| RCV | Text | 13890 | 9260 | 781265 | 104 | Raw Text | DAG |
| NYT | Text | 175299 | 177067 | 180659 | 2109 | Raw Text | - |

Table 1: Statistics of the datasets used in the experiments.

that reduces energy defined by structured energy network; Energy network is only used as a tool for evaluation of loss and SEAL framework is not interested in mimicking the test-time behaviour of SPEN as INFNET tries to. Despite the difference in the end goal, mechanically speaking, INFNET is a special case of SEAL (i.e. SEAL with $L_{E-\text{margin}}$) if the margin is removed from the SSVM loss for the feedforward model udpate. In fact, Tu et al. (2020) reports that this special case leads to more efficient and stable learning of feedforward network in their ablation study. We use this special case of INFNET to represent SEAL with $L_{E-\text{margin}}$in the main experiment.

**Learning label embeddings for multi-label classification**    Utilizing label representation has been a popular approach for multi-label classification (Wang et al., 2018; Xiao et al., 2019; Zhang et al., 2021). These works embed labels as vectors, allowing for richer representations, and then utilizing cross-entropy loss which models individual label probability independently. While the energy network in SEAL represents an output probability, its main purpose is to allow the gradient for each label to depend on other label probabilities, a feature which is absent from these prior works, for which label interactions must be latently encoded in the embedding space.

**Eenrgy-based models used for reranking**    Recently, energy-based models (EBM) have been applied to text generation problems to first generate and rerank the generated sentences by EBM score (Deng et al., 2020; Bhattacharyya et al., 2021). The residual energy modeling of Deng et al. (2020) shares some similarity to the equation 9,apart from the minor difference that they utilize frozen language model for $P_N$ whereas SEAL has $P_N$ that constantly learns from the energy models. The major difference is that these previous works have simply used the scores of the energy network, whereas SEAL views the energy network as a loss function, leveraging the gradient from the energy network to further train another neural network.

## 4 EXPERIMENTS

In our experiments, we use 10 multi-label classification datasets (shown in Table 1) covering varied label space, size, as well as input characteristics. The first 7 are small and medium sized feature based datasets, while the last three Blurb Genre Collection (BGC), RCV (Reuters 1), and New York Times (NYT) are large datasets with raw text as the input.[4]

**Energy Network:** We use the same structure for the energy network $E_\Theta(x, y)$ as described in Belanger & Mccallum (2016), wherein the energy is the sum of local $E_\Theta^{\text{local}}(x, y)$ and global $E_\Theta^{\text{global}}(x, y)$ energies defined as:

$$E_\Theta^{\text{local}}(\mathbf{x}, \mathbf{y}) = \sum_{i=1}^{L} y_i \mathbf{b}_i^\top T_E(\mathbf{x}) , \qquad E_\Theta^{\text{global}}(\mathbf{x}, \mathbf{y}) = \mathbf{v}^\top \sigma(\mathbf{M}\mathbf{y}), \qquad (11)$$

where $\mathbf{b}_i, \mathbf{v}, \mathbf{M}, T_E$ contain learnable parameters, $\sigma(z) = \log(1 + e^z)$ is the softplus activation function, and $T_E$ is a feature network for structured energy network. Note that due to the presence of

---

[4]Due to their enormous size, we use a subset of NYT training set, which is still the largest dataset in our experiments.

Table 2: Performance of SEAL framework (row 5-9) compared to feedforward network trained with cross-entropy (row 1) and sturctured energy networks (row 2-4) that are learned with $L_E$ described in earlier section. We observe feedforward network learned with SEAL is almost always better than cross-entropy learned model. We also observe while $L_{E-\text{margin}}$ and $L_{E-\text{regression}}$ does not show significant difference, $L_{E-\text{NCEranking}}$ and $L_{E-\text{ranking}}$ has stronger performance in general.

| method | samples | discrete input datasets | | | | continuous input datasets | | |
|---|---|---|---|---|---|---|---|---|
| | | **bibtex** | **delicious** | **genbase** | **cal500** | **eurlexev** | **expr_fun** | **spo_fun** |
| cross-entropy | x | 42.40 | 30.16 | 47.37 | 33.58 | 42.19 | 37.48 | 27.97 |
| **energy only** | | | | | | | | |
| SPEN | x | **42.99** | 26.76 | 32.50 | 39.36 | **41.75** | **36.43** | 27.20 |
| DVN | x | 42.73 | **29.59** | **78.43** | **48.21** | 28.90 | 32.19 | **29.97** |
| NCE | o | 17.93 | 16.78 | 20.64 | 38.87 | 0.19 | 27.12 | 19.33 |
| **SEAL** | | | | | | | | |
| margin | x | 42.86 | 29.75 | 96.53 | 36.69 | 41.83 | 37.91 | 28.43 |
| regression | x | 43.74 | 29.79 | 96.95 | 37.97 | 41.65 | 38.12 | 28.89 |
| regression-s | o | 44.53 | 29.87 | 96.81 | 38.95 | 42.32 | 37.84 | 28.02 |
| NCEranking | o | **44.76** | 34.79 | **97.32** | **41.62** | **42.78** | **38.21** | 28.70 |
| ranking | o | 44.20 | **36.04** | 96.60 | 40.71 | **42.78** | 38.03 | **29.68** |

$E_\Theta^{\text{global}}$ term in $L_F$, the gradient for feedforward network, $\frac{\partial L_F}{y_i}$, can capture the dependency of $y_i$ to all $L$ dimensions of **y**. The expression for this is provided in Appendix B.

**Feature Network:** We use the same feature network structure for both the energy and feedforward network denoted as $T_E, T_F$ respectively. For the the feature-based datasets, the feature network $T : \mathbb{R}^d \to \mathbb{R}^h$ consists of multi-layer perceptron with with softplus nonlinearities. For the raw text based datasets, we use pre-trained BERT (Devlin et al., 2019) with adapter (Houlsby et al., 2019; Pfeiffer et al., 2020) as the feature network.

**Feedforward Network:** Given a feature network $T_F$ that generates features in $\mathbb{R}^h$, the feedforward network $F_\Phi(x) = \mathbf{G}T_F(x)$, where $\mathbf{G} \in \mathbb{R}^{L \times h}$ is a matrix consisting of learnable embeddings, one row of embedding for each label.

**Training:** The models for feature-based datasets are trained on a single TitanX GPU with 14GB CPU memory and the models for BCG, RCV and NYT are trained on m40 GPU. We use separte ADAM optimizer (Kingma & Ba, 2014) for energy network ($\Theta$) and for feedforward network ($\Phi$) which optimizes parameters in alternating fashion[5]. For each minimization loop, we take $n_E, n_F$ gradient steps respectively for estimating $\Theta_t$ and $\Phi_t$. We leave these $n_E, n_F$ as hyperparameters to be tuned. The best hyper-parameters for each model are found using Bayesian search[6] and reported in Appendix D. We also report train, inference time and parameter size of different methods in AppendixE.

## 4.1 RESULTS FOR FEATURE BASED DATASETS

The experiment results across 7 feature-based dataset are presented in Table 2. The table consists of feedforward network only trained with cross-entropy (row 1), energy networks (row 2-4) described in previous section that is evaluated with GBI, and lastly feedforward network trained with SEAL framework with different types losses (row 5-9).

SEAL outperforms cross-entropy loss (CE) on almost all datasets with average gains over CE ranging from +0.6 (SEAL-$L_{E-\text{margin}}$) to +2.85 (SEAL-$L_{E-\text{NCEranking}}$) F1 points. We excluded excessive gain of genbase in computing the average as we believe it is an outlier. Further analysis on the performance on genbase is provided in Appendix F.

---

[5]The code we used to train and evaluate our models is available a `https://anonymous.4open.science/r/SEAL/README.md`.

[6]We use Weights & Biases (Biewald, 2020) for hyperparameter search

Between $L_E$ types, $L_{E-\text{NCEranking}}$ and $L_{E-\text{ranking}}$ performs the best while $L_{E-\text{margin}}$ and $L_{E-\text{regression}}$ performs similarly on average. $L_{E-\text{NCEranking}}$ and $L_{E-\text{ranking}}$ are the only ones that always outperforms CE in all 7 datasets whereas the others sometimes perform slightly worse than CE.

With the high gains of $L_{E-\text{NCEranking}}$ and $L_{E-\text{ranking}}$, we are likely to conclude using samples rather than a single point can capture the output surface of the feedforward network better. To examine this hypothesis in more detail, we test sampling approaches on regression-based loss as well. We define $L_{E-\text{regression-s}} = \sum_{\tilde{\mathbf{y}} \in S} L_{E-\text{regression}}(\mathbf{x}, \mathbf{y}^*, \tilde{\mathbf{y}}; \Theta)$ given sample set $S$. We take two approaches in collecting these samples: 1. discrete binary vectors drawn from probability vector $\tilde{\mathbf{y}}$ as done in $L_{E-\text{NCEranking}}$ and 2. continuous perturbation of $\tilde{\mathbf{y}}$ with gaussian noise. We found that discrete samples did not make any notable changes between $L_{E-\text{regression}}$ and $L_{E-\text{regression-s}}$, however, found that using continuous samples made +0.4 F1 improvements in average and report $L_{E-\text{regression-s}}$ as the version which uses continuous sample. We conlcude that using samples are helpful in capturing better energy surface for traininig feedforward network, however, effect and characteristics of sample might differ per types of $L_E$. We believe the sampling is more effective on $L_{E-\text{NCEranking}}$, $L_{E-\text{ranking}}$ as groups of samples contribute in relative manner in learning surface in contrast to $L_{E-\text{regression}}$.

To further study effect of different $L_E$, we examine whether there is a correlation in performance between the isolated structured energy network and SEAL which utilizes identical $L_E$. For training isolated structured energy network with margin-based and regression-based energy loss, we follow training approaches of SPEN (Belanger & McCallum, 2016) and DVN (Gygli et al., 2017). For NCE, we need access to noise distribution $P_N$. we set pretrained feedforward network probability $P(Y|X; \Phi)$ as $P_N(Y)$. We do not see much of correlation between the two, in other words, characteristics of $L_E$ *that trains best structured energy network* and $L_E$ *that trains best feedforward network* are different. While NCE seems to perform worst when utilizing $L_{E-\text{NCEranking}}$ in isolated manner, the constrastive characteristic of $L_{E-\text{NCEranking}}$ and $L_{E-\text{ranking}}$ that is tailored to feedforward network seems to be helpful. Regardless of $L_E$ types, we also obesreve that it is much more stable and usually higher-performing to train with SEAL than to train energy network separately. In Appendix C, we also show compare individual energy network as a prediction model compared to SEAL in detail.

## 4.2 RESULTS FOR LARGE TEXT DATASETS

Table 3: Test F1 for text datasets.

| method \ datasets | BGC | RCV | NYT |
|---|---|---|---|
| cross-entropy | 81.15 | 87.18 | 77.4 |
| SEAL-margin | 81.14 | 87.01 | 78.13 |
| SEAL-NCEranking | **81.64** | **87.82** | **78.87** |

In order to test the effectiveness of SEAL in the scope of large pre-trained models, we examine text datasets with pre-trained BERT (Devlin et al., 2019) with adapter (Houlsby et al., 2019; Pfeiffer et al., 2020) as the feature network $T_E$ and $T_F$ as shown in Table 3. Considering the computational expense of running hyper-parameter searches on BERT, based on Table 2, we choose the best-performing SEAL (NCEranking) to compare with baselines: SEAL-margin and cross-entropy. The NYT subset we used has almost 17x larger training data than the largest feature-based data and SEAL framework outperforms cross-entropy fine-tuning. Overall, the Table 3 shows that SEAL, especially SEAL-NCEranking, is effective on pre-trained models that utilizes large text datasets as well. On average, SEAL-NCEranking gains 0.87, 0.68 F1 points over cross entropy and SEAL-margin, respectively, on text datasets as shown in Table 3.

## 5 FURTHER ANALYSIS

**SEAL without cross entropy:** Figure 2a shows the performance of SEAL when the cross-entropy loss (CE) is completely removed from equation 1. We first observe that surprisingly that SEAL itself without resorting to CE learns a model that sometimes outperforms model trained with CE ( cal500 and spo_fun). However, we also observe that SEAL without CE is unstable as performance varies a lot across different datasets. This tells us that SEAL requires a signal that can bring the model $F_\Phi(\cdot)$ into reasonable region first in order to demonstrate stable performance shown in Table 2. Another

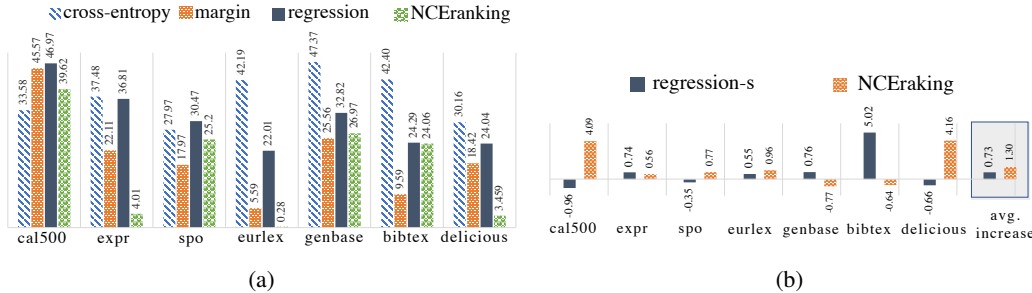

Figure 2: (a) The performance of SEAL variants without cross-entropy loss term on $L_F$, i.e. when $\lambda_1 = 1, \lambda_2 = 0$ in equation 1, compared to the model that only learns with cross-entropy. (b) Change in test F1 when moving from SEAL-static to SEAL-dynamic.

interesting observation is that SEAL-NCEranking model when it is often the weakest without CE can give the largest improvement when combined with CE together in SEAL as shown in Table 2. In short, we find CE to bring a stable synergy with energy network loss within SEAL: surpassing each individual's performance significantly when combined together. While there could be multiple techniques to stabilize learning of SEAL besides adding CE loss, such as pre-training techniques, we leave those interesting studies as future work.

**Effect of dynamic loss computation:** We now compare the effect of dyanmic loss learning by comparing SEAL to SEAL-static in Figure 2b We observe that pretrained energy networks with SEAL-static also results in competitve results, often better performing than the simple cross-entropy loss. However, we oberve that in average, the dynamic loss learnig of SEAL outperforms SEAL-static for $L_{E-\text{regression}}$, and $L_{E-\text{NCEranking}}$.

**Effect of applying ranking loss directly on $F_\Phi$** With the large improvement that $L_{E-\text{NCEranking}}$ and $L_{E-\text{ranking}}$ has, we examine whether the gain is actually coming from SEAL framework or simply from the power of ranking loss itself which can be applied to feedforward network $F_\Phi$ directly. We conduct this ablation study over relatively small datasets: genbase, cal500 and delicious. While we observed a near +10 F1 point in genbase, we saw decrease in performance in cal500 (-1.0 F1) and delicious (-2.3 F1) compared to plain cross-entropy model. With these experiments, we notice two trends. First, we again observe that genbase is an outlier where capturing structure can greatly increase the performance although not as much as SEAL framework helped. Two, we confirm that the ranking loss without energy network is not very effective in training feedforward model $F_\Phi$. This again confirms that the energy network $E_\Theta$ plays a major role to train $F_\Phi$ in the SEAL framework.

## 6 CONCLUSION

We propose SEAL: a framework that can adopt structured energy network as a trainable loss function for training feedforward network. Through examining different energy losses that trains energy network, we show that SEAL is a general framework which is effective over different loss functions and different architectures, such as MLP and BERT-based adapter. Through extensive result on 7 feature-based and 3 text-based datasets, we show that SEAL brings synergy between isolated energy newtork and cross entropy function: performing better and in more stable manner when combined together. Lastly, through ablation study, we examine the benefit of dyanmic loss learing which in average brings a gain over static version of SEAL, i.e. SEAL-static. This research opens up doors for various future work, including but not limited to, application of learned energy loss in unsupervised dataset and exploring different architectures of energy network. In the long horizon, the authors are interested in whether general pretrained scoring neural networks, such as BLEURT (Sellam et al., 2020) and BERT-score (Zhang et al., 2019), could serve as a loss function when it can provide a backpropagatable gradients to the output space.

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

APPENDIX

## A  NCE RANKING METHOD WITH $s(\mathbf{x}, \mathbf{y}; \Theta) = -E_\Theta(\mathbf{x}, \mathbf{y})$

As pointed out by Ma & Collins (2018) as well, If we set $s(\mathbf{x}, \mathbf{y}; \Theta) = -E_\Theta(\mathbf{x}, \mathbf{y})$, we can regard this this learning with $E'_\Theta(\mathbf{x}, \mathbf{y}) = E_\Theta(\mathbf{x}, \mathbf{y}) + \log P_\mathrm{N}(\mathbf{y})$. As pointed out earlier, this leads the model to estimate $P(\mathbf{y}|\mathbf{x})$ with

$$\frac{\exp\left(-E'_\Theta(x, y)\right)}{\sum_{y \in \mathcal{Y}} \exp\left(-E'_\Theta(x, y)\right)}.$$

Plugging in $E_\Theta'$ as $E_\Theta(\mathbf{x}, \mathbf{y}) + \log P_\mathrm{N}(\mathbf{y})$ once again, we can see that $P(Y|X)$ is estimated by

$$\frac{P_N(\mathbf{y}) \exp\left(-E_\Theta(\mathbf{x}, \mathbf{y})\right)}{\sum_{\mathbf{y} \in \mathcal{Y}} P_N(\mathbf{y}) \exp\left(-E_\Theta(\mathbf{x}, \mathbf{y})\right)}$$

## B  GRADIENT TO THE LABEL REPRESENTATIONS

As described in section 4, the energy component of loss $\mathrm{L}_F$, for the feedforward network, is the sum of following two parts:

$$E_\Theta^{\mathrm{local}}(\mathbf{x}, \mathbf{y}) = \sum_{i=1}^{L} y_i \mathbf{b}_i^\top T_E(\mathbf{x}) \quad \text{and} \quad E_\Theta^{\mathrm{global}}(\mathbf{x}, \mathbf{y}) = \mathbf{v}^\top \sigma(\mathbf{My}). \tag{12}$$

The gradient of $E^{\mathrm{global}}$ w.r.t $\mathbf{y}$ is given by the following vector of length $L$

$$\frac{\partial E^{\mathrm{global}}}{\partial \mathbf{y}} = \mathbf{M}^T \mathrm{Diag}(\sigma'(M\mathbf{y}))\mathbf{v}. \tag{13}$$

While the gradient of cross entropy $l(y, y^*)$ w.r.t. $\mathbf{y}$ is given by the following vector of length $L$

$$\frac{\partial l}{\partial \mathbf{y}} = \mathbf{y}^* \otimes \frac{1}{\mathbf{y}} - (1 - \mathbf{y}^*) \otimes \frac{1}{1 - \mathbf{y}}, \tag{14}$$

where $\otimes$ is element-wise product.

Note the dependency of the gradient for a particular component $y_i$ of $y$ on other component in the expressions above. The gradient of cross entropy ($\frac{\partial l}{\partial y_i}$) only depends on its label dimension $i$: $y_i$ and $y_i^*$. In contrast, in the case of the energy function as loss (SEAL), the gradient $\frac{\partial \mathrm{L}_F}{\partial y_i}$ depends on all $L$ dimensions of $\mathbf{y}$ due to the product term $\mathbf{My}$ in $\frac{\partial E^{\mathrm{global}}}{\partial y_i}$.

## C  COMPARISON OF SEAL AND SEPARATE ENERGY NETWORK

In the following figures, we compare the F1 scores of SEAL framework with $\mathrm{L}_{E-\mathrm{regression}}$ (Figure 3), $\mathrm{L}_{E-\mathrm{NCEranking}}$ (Figure 4) and $\mathrm{L}_{E-\mathrm{margin}}$ (Figure 5) on validation set with its energy-only counterpart. We further report validation F1 scores for the output of GBI starting from the samples drawn from $\hat{y}$ and randomly. We show this comparison to evaluate the performance of the energy network on points around $\hat{y}$ and random points.

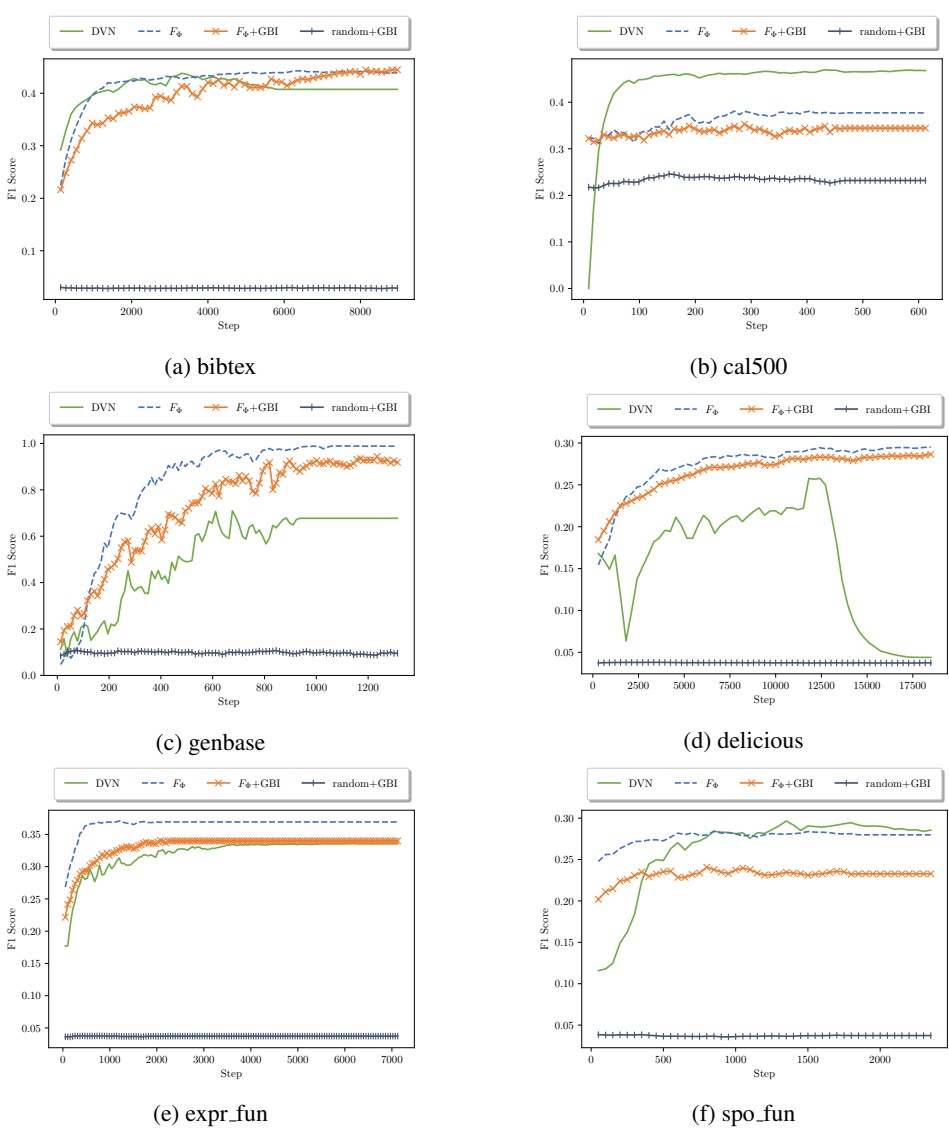

(a) bibtex

(b) cal500

(c) genbase

(d) delicious

(e) expr_fun

(f) spo_fun

Figure 3: Validation F1 performance of SEAL framework with $L_{E-\text{regression}}(F_\Phi)$ compared to energy-only (DVN). Further, we compare validation F1 performance of GBI starting from samples drawn from feedforward network's output ($F_\Phi$+GBI) and samples drawn randomly (random + GBI).
.

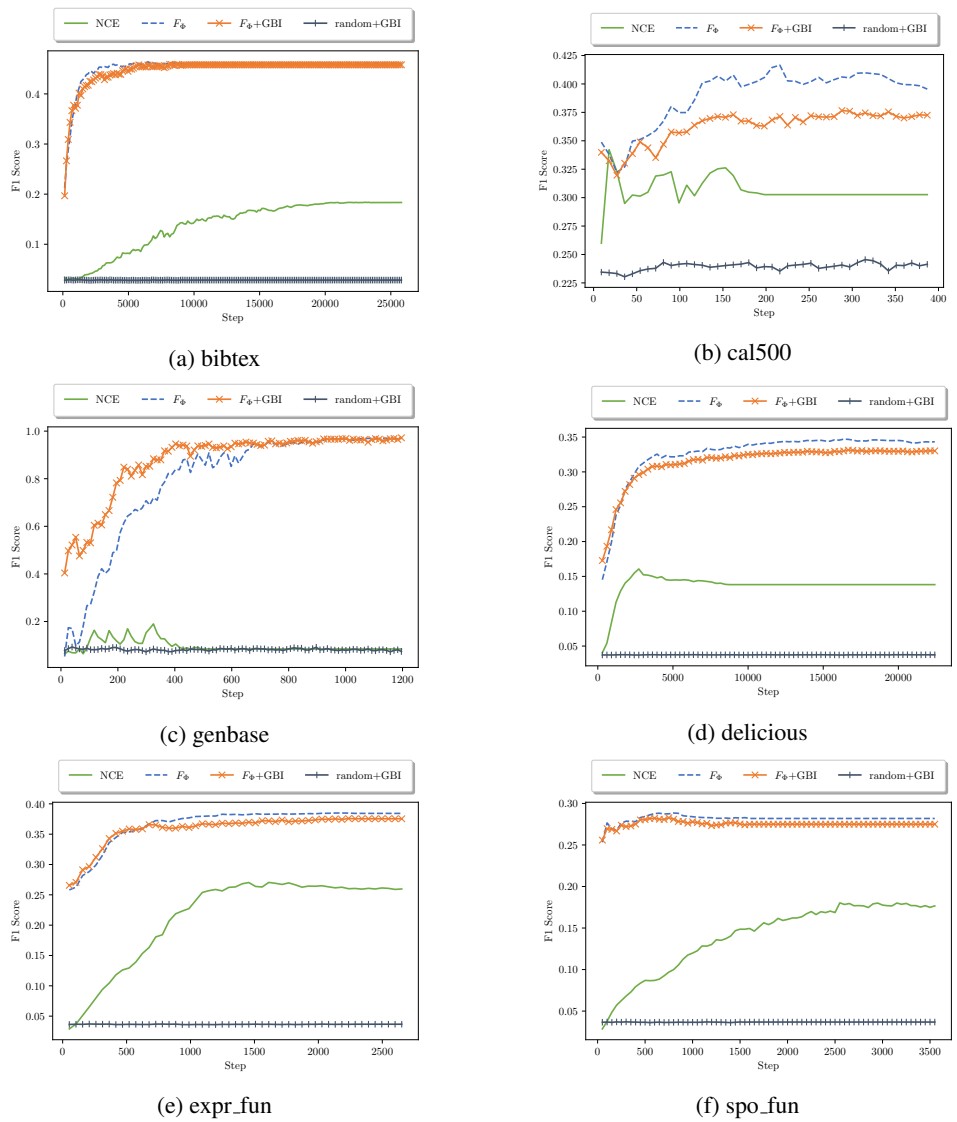

Figure 4: Validation F1 performance of SEAL framework with $L_{E-\text{NCEranking}}(F_\Phi)$ compared to energy-only (NCE). Further, we compare validation F1 performance of GBI starting from samples drawn from feedforward network's output ($F_\Phi$+GBI) and samples drawn randomly (random + GBI).

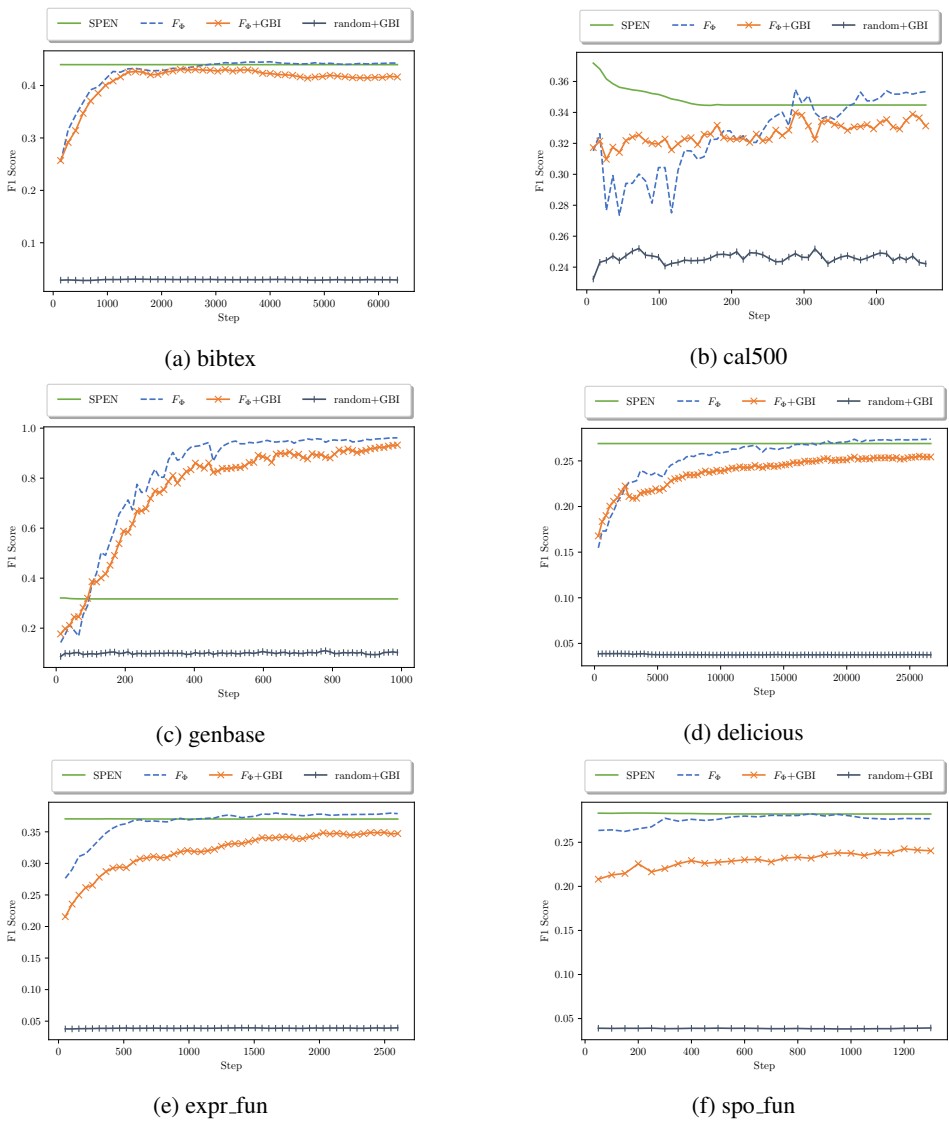

(a) bibtex

(b) cal500

(c) genbase

(d) delicious

(e) expr_fun

(f) spo_fun

Figure 5: Validation F1 performance of SEAL framework with $L_{E-\text{margin}}(F_\Phi)$ compared to energy-only (SPEN). Further, we compare validation F1 performance of GBI starting from samples drawn from feedforward network's output ($F_\Phi$+GBI) and samples drawn randomly (random + GBI).

## D HYPERPARAMETERS

The following tables tabulate the hyper-parameters for SEAL framework for each dataset. We further fix the number of layers, layer-dimensions and dropout in feed-forward network based on the configuration of the best model trained using cross-entropy. We fix the $\lambda_2 = 1$ in equation 1 to reduce degree of freedom in hyperparameter search.

| method | $\lambda_1$ | num samples | $\Theta_{lr}$ | $\Phi_{lr}$ |
|---|---|---|---|---|
| margin | 0.001 | n/a | 0.0002 | 0.001 |
| regression | 0.002 | n/a | 0.0003 | 0.001 |
| regression-s | 0.03 | 20 | 0.0004 | 0.001 |
| NCEranking | 0.002 | 100 | 0.0005 | 0.001 |
| ranking | 0.002 | 100 | 0.000015 | 0.001 |

Table 4: genbase

| method | $\lambda_1$ | num samples | $\Theta_{lr}$ | $\Phi_{lr}$ |
|---|---|---|---|---|
| margin | 0.001 | n/a | 0.0005 | 0.008 |
| regression | 1.4 | n/a | 0.006 | 0.007 |
| regression-s | 9 | 5 | 0.008 | 0.006 |
| NCEranking | 0.4 | 40 | 0.007 | 0.005 |
| ranking | 8 | 80 | 0.003 | 0.0001 |

Table 5: cal500

| method | $\lambda_1$ | num samples | $\Theta_{lr}$ | $\Phi_{lr}$ |
|---|---|---|---|---|
| margin | 0.0002 | n/a | 0.002 | 0.004 |
| regression | 8 | n/a | 0.0001 | 0.00015 |
| regression-s | 9 | 5 | 0.0001 | 0.003 |
| NCEranking | 0.9 | 20 | 0.002 | 0.001 |
| ranking | 1 | 60 | 0.0008 | 0.0001 |

Table 6: delicious

| method | $\lambda_1$ | num samples | $\Theta_{lr}$ | $\Phi_{lr}$ |
|---|---|---|---|---|
| margin | 0.001 | n/a | 0.0003 | 0.002 |
| regression | 0.002 | n/a | 0.0006 | 0.004 |
| regression-s | 0.01 | 20 | 0.0004 | 0.007 |
| NCEranking | 0.5 | 40 | 0.001 | 0.002 |
| ranking | 0.3 | 40 | 0.0003 | 0.002 |

Table 7: eurlex

| method | $\lambda_1$ | num samples | $\Theta_{lr}$ | $\Phi_{lr}$ |
|---|---|---|---|---|
| margin | 0.01 | n/a | 0.00003 | 0.001 |
| regression | 0.02 | n/a | 0.0003 | 0.0006 |
| regression-s | 2 | 30 | 0.007 | 0.002 |
| NCEranking | 4 | 80 | 0.00005 | 0.0002 |
| ranking | 1 | 40 | 0.008 | 0.0005 |

Table 8: expr_fun

| method | $\lambda_1$ | num samples | $\Theta_{lr}$ | $\Phi_{lr}$ |
|---|---|---|---|---|
| margin | 0.08 | n/a | 0.0001 | 0.002 |
| regression | 9 | n/a | 0.001 | 0.00015 |
| regression-s | 0.02 | 10 | 0.0002 | 0.0009 |
| NCEranking | 0.7 | 60 | 0.008 | 0.0005 |
| ranking | 1.5 | 40 | 0.004 | 0.002 |

Table 9: spo_fun

| method | $\lambda_1$ | num samples | $\Theta_{lr}$ | $\Phi_{lr}$ |
|---|---|---|---|---|
| margin | 0.0002 | n/a | 0.00005 | 0.001 |
| regression | 3 | n/a | 0.005 | 0.003 |
| regression-s | 9 | 10 | 0.005 | 0.001 |
| NCEranking | 5 | 20 | 0.0004 | 0.001 |
| ranking | 9 | 40 | 0.00005 | 0.001 |

Table 10: bibtex

# E  ANALYSIS ON PARAMETER SIZE AND SPEED OF INFERENCE AND TRAINING

Here, we present the analysis on number of parameters (Table 11), inference speed (Table 12)), and training speed (Table 13) that different methods utilize per dataset.

In summary, SEAL requires approximately twice number of parameters compared to cross-entropy loss on training time, however it requires the same amount of parameter on inference time. Similarly, train time (sec) per epoch of SEAL is slower than that of cross-entropy and energy network as SEAL requires multiple backpropagation steps for both energy network and feedforward network. Nonetheless, again, at inference time, the speed of CE and SEAL is the same and is much faster (2x-7x) than the inference of energy network.

The analysis we present below are on the seven feature-based dataset, however the trend is similar for text-based dataset as well: twice parameter size on training time but equal parameter and runtime in inference time with CE method. The only difference in the text-based MLC is, due to running on a much larger BERT-baed models, that we only run one step for each update of feedforward network and energy network. Thus, unlike Table 13, train time of SEAL is approximately twice that of CE.

| Dataset \ Methods | Parameter size | | | |
|---|---|---|---|---|
| | CE | Energy network (SPEN, DVN, NCE) | SEAL (Train-time) | SEAL (Inference-time) |
| EXPR_FUN | 333000 | 533800 | 866800 | 333000 |
| SPO_FUN | 447000 | 597600 | 1044600 | 447000 |
| Bibtex | 958800 | 991000 | 1949800 | 958800 |
| Cal500 | 1158700 | 1123500 | 2282200 | 1158700 |
| Delicious | 754000 | 951000 | 1705000 | 754000 |
| Genbase | 485600 | 491400 | 977000 | 485600 |
| Eurlex-ev | 4747500 | 5546500 | 10294000 | 4747500 |

Table 11: The number of parameters required during train time for SEAL is approximately double the size of feedforward (CE column) while energy network and feedforward sizes are comparable. However, in the inference time, SEAL has an equal amount of parameters to the CE column as only feedforward network is utilized during inference time.

| Dataset | Inference time (sec) | | Inference speed (examples/sec) | | Speed ratio (CE and SEAL/ EnergyNetwork) |
|---------|---------------|-------------|---------------|-------------|---------------|
| | Energy network | CE and SEAL | Energy network | CE and SEAL | |
| EXPR_FUN | 1.33 | 0.22 | 638 | 3801 | 5.96 |
| SPO_FUN | 1.18 | 0.16 | 709 | 5231 | 7.38 |
| Bibtex | 3.6 | 1.78 | 414 | 840 | 2.03 |
| Cal500 | 0.24 | 0.10 | 438 | 1080 | 2.47 |
| Delicious | 5.35 | 1.39 | 599 | 2307 | 3.85 |
| Genbase | 0.23 | 0.13 | 574 | 1005 | 1.75 |
| Eurlex-ev | 24 | 12.22 | 162 | 317 | 1.96 |

Table 12: We simply average inference time for CE and SEAL variants as they are very similar. Likewise, we average the inference time of different energy networks. Here, inference time (sec) is recorded for the whole validation set. We also present speed per sec (example/sec) and speed ratio. The last column shows that CE and SEAL methods are 2x-7x faster in inference time than the energy networks.

| Methods\ Datasets | Train time (sec/epoch) | | | | | | |
|---------|---------|---------|--------|--------|-----------|---------|-----------|
| | EXPRFUN | SPO_FUN | Bibtex | Cal500 | Delicious | Genbase | Eurlex-ev |
| CE | 6.29 | 5.46 | 22.67 | 1.00 | 33.01 | 2.12 | 114.84 |
| SPEN | 10.99 | 11.01 | 28.04 | 2.53 | 37.31 | 3.57 | 136.10 |
| DVN | 10.95 | 10.02 | 32.12 | 1.87 | 55.24 | 2.79 | 129.94 |
| NCE | 3.71 | 3.83 | 13.97 | 2.82 | 22.03 | 3.68 | 89.33 |
| SEAL-margin | 27.96 | 35.24 | 212.94 | 5.78 | 44.60 | 8.73 | 260.10 |
| SEAL-regression | 46.32 | 56.97 | 27.78 | 8.07 | 143.90 | 13.91 | 352.63 |
| SEAL-regression-s | 42.87 | 77.92 | 179.97 | 8.48 | 45.33 | 10.79 | 221.33 |
| SEAL-NCEranking | 72.43 | 40.34 | 131.73 | 16.58 | 158.97 | 17.77 | 431.92 |
| SEAL-Ranking | 41.24 | 26.83 | 218.37 | 7.04 | 317.63 | 10.74 | 408.79 |

Table 13: The training time per epoch is presented per dataset and per loss function used. Due to different gpu types and node status, there are some outliers. Furthermore, as SEAL runs multiple number of backpropagation steps for energy network and feedforward network in the alternating optimization, direct comparison of training time is not really available. However, the general trend in terms of training time per epoch is CE < Energy Networks < SEAL.

# F DATA-SPECIFIC ANALYSIS

In this section, we analyze why certain datasets behaves differently.

## F.1 GENBASE

Here, we analyze the factors behind SEAL achieving almost perfect F1 score on the genbase dataset. It seems genbase stands out with its small label size (27 which is the smallest among Table 1) and with very clear pattern in the label space. Upon analysis, out of 27 labels, we found that 6 labels only occur as a singleton (by itself), 10 labels only occur as non-singleton, and 7 labels do not participate at all. This leaves only 4 labels occurring by themselves as well as with others. Not only that, only 7% (35/500) of training instances have more than one active label. In this peculiar setting, we conjecture feedforward networks with cross-entropy (CE) loss can easily learn singletons, and energy networks would easily learn co-occurrence, but learning both might be confusing for these two models at the opposite end of the spectrum. We believe the synergy of CE and a loss from the energy network enables capturing the best of both worlds and achieving a near-perfect score that none of the approaches by itself achieves. As can be seen in Table 2, neither cross-entropy nor energy networks are nearly as strong as SEAL methods.

## F.2 DELICIOUS AND CAL500

In Table 2, it seems SEAL-NCEranking and SEAL-Ranking are particularly well performing on cal500 and delicious. It turns out that cal500 and delicious have a very high diversity of 1 and 0.981 (Reference: MLC data repo). Diversity of 1 means that each data point holds a unique label set. It seems that the exposure to multiple samples and evaluations of their relative scores in NCEranking loss function can be helpful in a high-diversity setting.

