# OpenReview forum: "Structured Energy Network as a dynamic loss function. Case study. A case study with multi-label Classification"
_ICLR.cc/2022/Conference — ICLR 2022 Submitted_

### Official Review · Reviewer_8Y6b · 2021-11-03

**Correctness:** 2
**Technical Novelty And Significance:** 2
**Empirical Novelty And Significance:** 2
**Recommendation:** 3
**Confidence:** 4

**Main Review:**

The paper proposes an interesting problem -- capturing the higher level dependencies in data utilizing an energy function compared to assuming conditionally independent predictions.

However I have several major concerns about the paper.

The first major concern is that the underlying presentation of the paper is somewhat poor. For example, one of the major claims in the paper are not supported. The authors states such that an energy function is interpretted as a dynamic loss function, but the method section does not really specify what this means.

The paper further appears to be in a rushed state and there are a large number of different typos -- for example

noise contarstive estimation -> noise contrastive estimation.
hyptothesize -> hypothesize.
avergae -> average.
f1 -> F1.

In addition, the empirical comparisons appear to be limited and evaluated only on low dimensional datasets. Can the authors show a practical use case for the proposed approach on an existing large scale benchmark? Or compare utilizing existing datasets from past works?

The underlying technical novelty of the approach also seems a bit poor -- the main contribution appears to be applying an InfoNCE loss to train an energy function. It is also not clear from the text why this loss is beneficial during training. For example, the authors claim that their new loss prevents adversarial training. I don't follow this as both models are still optimized at the same time -- with one model encourage to have lower energy generations and another to increase the energy of generated samples.  For furthermore, why is it that the proposed loss would leave to a gain of performance?

**Summary Of The Paper:**

This paper proposes a method to utilize a SPEN as a trainable loss function for training a neural network. In particular, the authors propose to train the energy function utilizing NCE loss compared to other loss functions used by prior works, and show that such a choice improves performance of the resultant classifier.



**Summary Of The Review:**

Due to poor presentation, a lack of empirical comparison, and no theoretical analysis in the underlying paper, I believe this paper is not in a suitable form for publication right now.

---

> ### Author Response · Authors · 2021-11-21
> **Response to weaknesses 1 and 2.  (W1,2 out of W1-4)**
>
> Thank you for taking your time and providing feedback. We identified four weakness-related comments (W1-4) that the reviewer has pointed out. We split the answer into two threads for the ease of following up on each topic: weaknesses 1 and 2 (W1,2) in one thread and weaknesses 3 and 4 (W3,4) in another thread.
>
> > W1: "The first major concern is that the underlying presentation of the paper is somewhat poor. For example, one of the major claims in the paper are not supported. The authors states such that an energy function is interpretted as a dynamic loss function, but the method section does not really specify what this means."
>
> We clearly state the definition of our dynamic version on p.3 of the method section: “We finally present SEAL which learns Θ^t  dynamically at step t” right above equation (4),(5) which defines the dynamic loop. We are open to criticism, however other reviewers (vt8f, CpcQ) explicitly mention that the paper is clearly written and highlight that figure 1 was helpful in delivering the concept of dynamic loss learning. Please let us know, in detail, what exactly confused you, and we will try our best to address it.
>
> > W2: For example, the authors claim that their new loss prevents adversarial training. I don't follow this as both models are still optimized at the same time -- with one model encourage to have lower energy generations and another to increase the energy of generated samples.
>
> First of all, we want to make a clarification that **we do not make a claim that we prevent adversarial training** per the reviewer’s comment. **We simply do not have an adversarial setup as energy loss $L_E$ and $L_F$ are independent**. $L_E$ learns to locate low energy to true input and the energy term in $L_F$ updates feedforward parameters ($\Phi$) to output low-energy outputs, i.e. Reducing $L_F$ does not necessarily increase $L_E$ and vice versa.
>
> To elaborate, the adversarial setup of INFNET  can be expressed as following:
> $\min_{\Theta} \max_{\Phi}  L_E (x,y,F_{\Phi}(x); \Theta)$
> where $\tilde{\mathbf{y}}$ in $L_E (\mathbf{x},\mathbf{y}, \tilde{\mathbf{y}}; \Theta)$ is replaced with $F_{\Phi}(x)$. In contrast, in the case of the SEAL, energy loss $L_E$ and $L_F$ are completely separated.
>
> While the above explanation suffices in clarifying the adversarial vs. non-adversarial framework, given this opportunity, we want to stress that this difference occurs from the difference in fundamental goal. As described in section 3, **the focus of all previous work (SPEN, DVN, INFNET) was on training the best possible energy network**. They achieve this goal by finding an example that could maximize the learning experience of the energy network -- by identifying an example that has the largest loss (i.e. adversarial). Describing a bit further, the goal of INFNET (Tu et al., 2019 & Tu et al., 2020), as described in their paper, was mimicking the inference-time behavior of this well-trained energy network.
>
> In contrast, **the focus of SEAL is in providing the best learning signals to the feedforward network**. To do so, the goal of the energy network in SEAL is *capturing the energy surface focused on output distribution of the feedforward network* rather than learning a good energy surface *on every space*. Therefore, SEAL just learns the energy network on the outputs of the feedforward network directly without requiring adversarial samples. SEAL is not interested in mimicking the inference-time behavior of the energy network; it is rather interested in utilizing the energy network as an evaluator that can provide useful learning signals to the feedforward network.

---

> ### Author Response · Authors · 2021-11-21
> **Response to weaknesses 3 and 4. (W3,4 out of W1-4)**
>
> Thank you for taking your time and providing feedback. We identified four weakness-related comments (W1-4) that the reviewer has pointed out. We split the answer into two threads for the ease of following up on each topic: weaknesses 1 and 2 (W1,2) in one thread and weaknesses 3 and 4 (W3,4) in another thread.
>
> > W3: (The novelty of NCE method) The underlying technical novelty of the approach also seems a bit poor -- the main contribution appears to be applying an InfoNCE loss to train an energy function.
>
> **In our experiment, NCE loss is not very helpful on its own without the SEAL framework,  neither for energy network nor for feedforward network.** As it can be seen in the experiments of training energy networks with NCE (Table 2 NCE), it performs very poorly when you apply gradient-based inference on the model trained with NCE. As shown in the ablation study (Sec 5. *Effect of applying ranking loss directly on $F_\phi$*), the ranking loss is also not very useful when it is directly applied to the feedforward network. In our experiments, the proposed SEAL-NCE was the only setup that gave a high boost in performance and we argue that this is a novel contribution.
>
> To elaborate further, given the SEAL framework where we have access to the feedforward network and energy network, the NCErankikng loss benefits from the feedforward network to gather negative samples, and energy network trained with NCEranking further teaches the feedforward network. To the best of our knowledge, we believe this structure was never explored before.
>
> **Q) Question to reviewer 8Y6b on this topic.**
> We did not understand the reviewer's reference to InfoNCE as opposed to the NCEranking paper we refer to [[Ma & Collins, 2018](https://arxiv.org/abs/1809.01812)]. Upon our review, InfoNCE seems closer to the "ranking" loss we used in equation (9), but do reviewers recommend InfoNCE to be closer to our literature than the other classical NCE methods? If so, could you perhaps teach us more about why you think our paper is closer to infoNCE?
>
> Also, we believe a direct comparison of InfoNCE would be directly using ranking loss to the feedforward network $F_{\Phi}$.  As we examine in our ablation study (Sec 5. *Effect of applying ranking loss directly on $F_\phi$*), this approach results in no significant difference to CE on our datasets. Again, NCE loss was not very helpful on its own without the SEAL framework in our experiments.
>
> > W4 (larger benchmark) Can the authors show a practical use case for the proposed approach on an existing large scale benchmark? Or compare utilizing existing datasets from past works?
>
>  We would like to point out that standard MLC is still considered a challenging problem in the machine learning community and is an area of active research. **SEAL is tested on three datasets with label cardinality equal greater than 1000: Delicious with 1k, NYT with 2k, and Eurlex with 4k**. This is a larger number of high-cardinality data sets compared to previous works (For more details, please refer to our response to reviewer CpcQ W1). Moreover, we believe evaluating on small datasets is as important as large labeled data is not always accessible in the real world, often referred to as a low-resource problem. With these facts in mind, we argue that our portfolio of experiments are more comprehensive than other MLC or energy network papers.
>
> Regarding the question  **utilizing existing datasets from past works**:  bibtex and delicious, which were utilized in the past work, have been included in our experiments for making comparisons with existing models like SPEN, DVN, and INFNET.
>
> Lastly, we examine two datasets previously utilized in the text MLC problem:  BGC and RCV. We argue that the level of performance are competitive and comparable to other papers. The capsule network on BGC performs 74.3 F1 [[Aly et al., 2019](https://aclanthology.org/P19-2045)](Table 2) whereas the BERT-adapter model with CE and SEAL-NCE has 81.15 F1 and 81.64 F1,  respectively.  [[Zhang et al., 2021](https://arxiv.org/abs/2106.03103)](Table 2) improves RCV F1 from 87.7 to 88.5, whereas our CE vs. SEAL-NCE gets 87.18 to 87.82. On the RCV dataset, due to train size* mismatch, there is a slight performance difference, but we argue that this is a competitive performance range to study the effect of the loss function.
>
> *On the RCV dataset, the slight performance difference might be from the fact that we are using an official training set of RCV1-V2 which is a significantly smaller training set (1.7%) compared to that of [[Zhang et al., 2021](https://arxiv.org/abs/2106.03103)].
>
> **References**
> * Aly et al., Hierarchical Multi-label Classification of Text with Capsule Networks, ACL 2019.
> * Zhang et al., Enhancing Label Correlation Feedback in Multi-Label Text Classification via Multi-Task Learning, ACL 2021 Findings.

---

> ### Comment · Reviewer_8Y6b · 2021-11-29
> **Rebuttal Response**
>
> I thank the authors for responding to rebuttal. The rebuttal somewhat addresses some of concerns I have (and I have improved my score slightly). However, I still found the paper somewhat difficult to read. Furthermore, I still believe this paper needs additional evaluation on larger datasets (as noted by Reviewer CpcQ) as well as comparison with BERT model with finetuning (noted by Reviewer CpcQ). Finally, the underlying method still appears somewhat incremental in nature.

---

> > ### Author Response · Authors · 2021-11-30
> > **Additional response to reviewer 8Y6b**
> >
> > Thank you for taking the time to go through our response and updating your rating.
> >
> > We kindly ask reviewer 8Y6b, whether the reviewer had a chance to read our response to reviewer CpcQ on [BERT+adapter vs. BERT](https://openreview.net/forum?id=dEOeQgQTyvt&noteId=q1ZbwI_-XOg).
> > To briefly summarize, full BERT-base cross-entropy fine-tuned models do not always outperform BERT-base + adapter models. This is also shown in recent adapter papers. Lastly, current adapter-based SEAL results are on par or better than full BERT-base fine-tuned with cross-entropy. We also argue that we present a more efficient way to utilize the SEAL framework with the large pre-trained models.
> >
> > Based on our response, reviewer CpcQ has increased their rating. Would you reconsider your position based on our [response on BERT-adapter](https://openreview.net/forum?id=dEOeQgQTyvt&noteId=q1ZbwI_-XOg)?
> >
> > Second, based on our small survey on other multi-label classification datasets, we argued that the datasets are large enough with larger cardinality on the label space than usual (on general response and responses to reviewer 8Y6b and CpcQ). If these comparisons were not convincing enough for reviewer 8Y6b, could the reviewer provide further details on why that is the case?
> >
> > Lastly, for clarity of writing, two reviewers (CpcQ and vt8f) all point out the clarity of writing as a strength of this paper as follows:
> > > **CpcQ**: The paper is well-structured, with clear relationships between each section. The writing is good.
> >
> > > **vt8f**: The paper is well written.
> > If reviewer 8Y6b still thinks our paper is hard to read, then could reviewer 8Y6b please point out which parts of the writing were hard to follow and suggest how we may improve the readability?
> >
> > We value the reviewer's time and again thank the reviewer for their feedback. We hope to hear from you soon again as the window for discussion is closing soon.

---

> > > ### Comment · Reviewer_8Y6b · 2021-11-30
> > > **Rebuttal Response**
> > >
> > > While I thank the author's for their response, I remain unconvinced. The approach itself still seems quite incremental to me, the writing itself appears unclear. In terms larger datasets, I do not mean larger label spaces, but rather on more established benchmarks. For example, what happens when this approach is applied to tasks in the GLUE benchmark? Or some other well established benchmarks?

---

> > > > ### Author Response · Authors · 2021-11-30
> > > > **Please provide suggestions for us to improve the paper.**
> > > >
> > > > We believe we are using well-established real world datasets in our experiments.
> > > > Feature-based datasets--bibtex and delicious--are used in numerous papers on MLC. Even recent works like DVN and InferenceNet, which are the baselines in this paper, used these datasets. For the text-based dataset, we also stated the performance of RCV and BGC that appeared in other papers for comparison in our response.
> > > > Even after providing the references of usage of these datasets in recent work, if the reviewer feels that these are non-standard datasets, we request the reviewer to provide at least one multi-label classification dataset that is different in some key characteristics compared to the datasets used in this paper and which the reviewer considers "well established".
> > > >
> > > > The reviewer suggests using the GLUE benchmark in our experiments. However, we do not think GLUE is a proper benchmark for this work because GLUE does not hold any MLC dataset, or for that matter, even a structured prediction dataset where multiple output variables interact. SEAL's intention is to capture the interaction between multiple output variables which is the main goal of structured prediction tasks. GLUE consists of tasks that are mostly binary or multi-class classification. We request the reviewer to provide a little more information regarding the following: **How should we evaluate an MLC model on binary or multi-class classification so as to assess its ability to perform MLC?**
> > > >
> > > > > the writing itself appears unclear.
> > > >
> > > > As stated in our previous response, we request the reviewer to provide some concrete steps that would make the writing clearer. In particular, what aspect of the writing do you find to be not clear?
> > > >
> > > > > The approach itself still seems quite incremental to me,
> > > >
> > > > Reviewer 8Y6B's original review stated that our method is not novel as we are just applying infoNCE. We have elaborated our reasoning for novelty before in our response and we asked why the reviewer thinks this is similar to InfoNCE. We haven't heard back on that matter, but to summarize again, InfoNCE does not have access to noise sample probabilities but only holds access to sampling from noise distribution; InfoNCE is similar to importance sampling which Ma & Collins (2018) state to be different with the proper NCE.
> > > > Not only that, we claimed that NCE loss itself did not help the model on its own; neither the energy network nor the feedforward network's performance is better than CE loss if NCE loss is applied without SEAL. With SEAL, the NCE loss showed the largest gains across all settings.
> > > >
> > > > We are not going to further elaborate on other novel aspects of this work as our previous response covers them. Instead, **as the reviewing process should be constructive and objective, we request the reviewer to provide suggestions or explanations** on how/why our work is incremental,  poorly written, and why our dataset is non-standard. We appreciate all criticism as it can lead to improving the paper. But improvement can only happen when the reasons for criticism are provided clearly. We would appreciate it if reviewer 8Y6B could provide explanations as to why our response does not address their concern.

---

> > > > > ### Comment · Reviewer_8Y6b · 2021-12-01
> > > > > **Rebuttal Response**
> > > > >
> > > > > In terms of novelty:
> > > > >
> > > > > This work takes the STEN work, but changes it so that both energy and predictive neural network are optimized jointly (as opposed to separately in the past). To me, this represents limited novelty compared to [Tu et al., 2020], as the only difference is to optimize both objective at the same time. Furthermore, there is no theoretical justification for why the underlying approach would work well -- the contribution of the paper is an empirical insight.
> > > > >
> > > > > Given the lack of theoretical grounding for the paper, and the rather incremental nature, it is important to show strong empirical benefits of the current technique. Despite this, the authors focus only on the task of multi-label classification (which prior works have focused on other tasks). I do not expect the authors to evaluate multi-label classification on something like the GLUE benchmark -- rather I would hope that they illustrate their approach can do something other than multi-label classification as this seems like a rather specific task for a somewhat incremental method.
> > > > >
> > > > > Finally, in terms of clarity, I found Section 2 to be somewhat hard to follow. In the first 2 paragraphs of Section 2, the
> > > > > author currently just introduce the SEAL framework, and the overall loss used to train both networks. However, the author don't provide a high level intuition on why they are doing what they are doing, and it may be good to provide
> > > > >  background on the idea of the energy landscape as a learnable loss function as well as intuition of what the energy function cover.  In subsequent paragraphs, the text makes statements such as: "instead, we hypothesize that it is more important to concentrate resource in depicting accurate energy surface around the current input, output pair", without providing exact detail on why the framework would allow more accurate energy surfaces. Finally, when introducing each of the separate energy training losses, it would be good to provide intuitions of the benefits of each different loss.
> > > > >
> > > > > There also appears to be a variety of separate grammatical issues which also subtract from clarity, some which I pointed out in the original submission of the paper. Some additional ones are below:
> > > > >
> > > > >  It would be useful to learn perfect energy surface, -> It would be useful to learn a perfect energy surface,
> > > > >  Instead, we hypothesize that it is more important to concentrate resource in depicting accurate energy surface around the ->   Instead, we hypothesize that it is more important to concentrate resources

---

### Official Review · Reviewer_CpcQ · 2021-11-03

**Correctness:** 3
**Technical Novelty And Significance:** 2
**Empirical Novelty And Significance:** 3
**Recommendation:** 6
**Confidence:** 4

**Main Review:**

The motivation of this paper is reasonable as the author pointed out in the introduction section: "The key idea is to provide the feed-forward network access to rich relationships in the output space through a learned loss function, and do this in a dynamic manner by updating the loss based on the current outputs of the feed-forward network".

General Significance: The main significance of the model is an adoption of STEN as a trainable loss function for learning an MLP, which could achieve better model quality with this combination. Regarding training stability and computation efficiency, from the discussion of the paper, it seems that back-propagation of gradient can be used in alternating fashion to optimizes the energy network (STEN) and the MLP. This seems easier and more stable (from the attached implementation based on AllenNLP library - pytorch) than training an energy network alone. The impact of this paper is limited because the experimental dataset are mostly on a label cardinality under 1000. It would be more interesting if the author could use a larger dataset (regarding number of train/val/test examples) and a larger label cardinality. Could take a look at Extreme datasets (http://manikvarma.org/downloads/XC/XMLRepository.html).

Novelty: This method is largely based on the STEN work (reused most building component of STEN, e.g., the energy network is built the same way), but it adds interesting extension which could open opportunities for the application of energy-based learning. Regarding related work, could you also discuss how "label embedding" (https://arxiv.org/abs/1503.08677) and "joint learning of label embedding and the feedforward network" (e.g., https://arxiv.org/pdf/1805.04174.pdf) relate to your proposed work here.

Technical Quality: The overall technical quality is on the fence. It would be nice if the author could provide some more explanations on the experiment results:
1. Why the ranking-based methods (in table 2) are significantly better than others in delicious and cal500 dataset?
2. Why the SEAL-based methods  lead by a large margin on genbase dataset? (is it something related to the nature that this is a very small dataset?)
3. Why DVN performs so well on cal500? (seems to be another small dataset).
4. What if you replace the adapter-based BERT encoder with a full finetuning on BERT?
5. Mentioned above, provide benchmark results on larger datasets.

Clarity: The paper is well-structured, with clear relationships between each section. The writing is good. I especially like Figure 1, which is very illustrative thats explains a good amount of details with a simple graph.

Minor typos (issues):
* Page 5. Section 3 Learning with dynamic loss. should probably say "Lastly, these models were designed for multi-class classification" (remove "do").


**Summary Of The Paper:**

This paper proposes a framework to use Structured Prediction Energy Network (SPEN, a prior work) as a loss function. Previously SPEN is a deep architecture to learn an energy function of candidate labels, which captures structural dependencies between labels that would lead to intractable graphical models. The author of this paper finds out that it is effective to jointly learn the parameters in SPEN and the parameters in the feed-forward network (MLP). In the training time, the model alternatingly minimizes the parameters in SPEN by fixing the parameters in MLP; and then minimizes the parameters in MLP by fixing the parameters in SPEN. Empirical results in 10 multi-label datasets are given, which shows good performance boost from using the proposed methods.

**Summary Of The Review:**

I am more inclined to marginally accept the paper given its current status.

---

> ### Author Response · Authors · 2021-11-19
> **Response (W1) to comments and questions of reviewer CpcQ**
>
> Thanks for your time and valuable feedback. In the comments above, we have identified two weakness-related comments (W1-2) and four questions (Q1-4) on analysis of experiment results and experiment setup. For ease of following each topic, we divide our answers into different threads: W1, W2, and Q1-4.
>
> > W1.  The impact of this paper is limited because the experimental dataset are mostly on a label cardinality under 1000. It would be more interesting if the author could use a larger dataset (regarding number of train/val/test examples) and a larger label cardinality. Could take a look at Extreme datasets (http://manikvarma.org/downloads/XC/XMLRepository.html).
>
> Due to the advent of big data, extreme MLC has garnered significant attention from the community recently. While extreme MLC holds different computational and scientific challenges compared to standard MLC, **we would like to point out that standard MLC is still considered as a challenging problem in the machine learning community and is an area of active research.** There have been several interesting works in the last couple of years that present methods for performing MLC for datasets with small to medium label-space cardinality. For instance,  [[Aly et al., 2019](https://aclanthology.org/P19-2045)], [[Pal et al., 2020](https://arxiv.org/pdf/2003.11644.pdf)] , [[Xiao et al., 2019](https://aclanthology.org/D19-1044/)], [[Zhang et al., 2021](https://arxiv.org/abs/2106.03103)] all  consider datasets with label cardinality ranging from 10 to 3000.
>
> **SEAL is tested on three datasets with label cardinality equal greater than 1000: Delicious with 1k, NYT with 2k, and Eurlex with 4k**. This is a larger number of high-cardinality data sets compared to previous works mentioned above. In terms of data (train/val/test) size, we also have datasets that are comparable to other MLC papers as most of them are in the order of 10k, 100k, with few exceptions. On the datasets we surveyed, there weren’t datasets that has both a large train instance and large label size at the same time. For example, [[Wang et al., 2018](https://arxiv.org/pdf/1805.04174.pdf)] recommended by the reviewer has large datasets with the size of (0.5m, 1m) but they only have very small label size (<15). Specifically, among the 12 datasets used in the 5 papers mentioned above, there was only one dataset that had a train set greater than 100k and label size greater than 1000 such as the NYT dataset in our paper.
>
> Some of the most commonly used small to medium datasets like bibtex and delicious have been included in our experiments for making comparisons with existing models like SPEN, DVN, and INFNET. Moreover, we believe evaluating on small datasets is as important as large labeled data is not always accessible in the real world, often referred to as a low-resource problem.
>
> We have carefully curated the portfolio of datasets used in our experiments--small-, medium-, and large datasets in terms of label cardinality and number of training examples, and various input feature characteristics (binary, continuous, text)--to show the general applicability of our method. **We evaluate SEAL on 10 datasets with a wide range of characteristics, resulting in a comprehensive empirical evaluation for the task of MLC, which is the focus of this paper.** Dismissing the impact of an MLC paper based on the fact that it is not performing extreme MLC is a little extreme in our opinion.
>
> **Reference**:
> * Aly et al., Hierarchical Multi-label Classification of Text with Capsule Networks, ACL 2019.
> * Pal et al, MAGNET: Multi-Label Text Classification using Attention-based Graph Neural Network, ICAART 2020
> * Wang et al., Joint learning of label embedding and the feedforward network, ACL 2018
> * Xiao et al., Label-specific document representation for multi-label text classification, EMNLP 2019.
> * Zhang et al., Enhancing Label Correlation Feedback in Multi-Label Text Classification via Multi-Task Learning, ACL 2021 Findings.

---

> ### Author Response · Authors · 2021-11-19
> **Response (W2) to comments and questions of reviewer CpcQ**
>
> Thanks for your time and valuable feedback. In the comments above, we have identified two weakness-related comments (W1-2) and four questions (Q1-4) on analysis of experiment results and experiment setup. For ease of following each topic, we divide our answers into different threads: W1, W2, and Q1-4.
>
> > W2. This method is largely based on the STEN work (reused most building component of STEN, e.g., the energy network is built the same way),
>
> Previously published papers on SPEN variants such as DVN, SG-SPEN all worked on the same network architecture for multi-label classification (MLC) and their novelty was in proposing different loss functions to train energy networks, not in proposing new architecture. In our work, we not only propose to use a new loss function (NCE) in application to the SPEN framework but propose a novel perspective of using the energy network *as a loss function*.
> The reason we chose an identical energy network structure was to make our proposed SEAL framework to be comparable to the previous work. We argue that we should not be penalized for controlling the orthogonal factors for a fair comparison.
>
> Furthermore, by proposing the SEAL framework, we argue that we removed SPEN’s weaknesses and only adopted the benefits in a novel way. SPEN has not been very popular as its inference and training steps are expensive as both steps utilize gradient-based inference (GBI) (Please refer to the runtime table in the response to vt8f). This GBI step is also highly dependent on hyperparameters such as step size, the number of steps, and the initial point to start GBI and this dependency brings variance in the performance of the model. In this paper, we rediscovered  SPENs as a loss function rather than its standard usage as a prediction network, which is a very general idea that has not been explored before. This proposal of SEAL removes the shortcoming of SPEN by removing the usage of GBI and only adopts its benefits by utilizing the fact energy networks can capture the whole structure of output probability vector $y$.
>
> Last, but not least, our proposal of NCE loss is motivated to uniquely benefit our proposed SEAL framework. As seen in Table 2, the energy network trained with NCE itself performs very bad and as seen in the ablation study, direct application of ranking loss to feedforward network does not bring much positive effect. Nonetheless, when combined with a feedforward network in the SEAL framework, it performs the best compared to all other setups. We argue that this proposal and discovery is a novel aspect of our work.
>
> In fact, comparing each isolated energy network and similar energy network used in the SEAL framework (i.e. SEPN vs. SEAL-margin, DVN vs. SEAL-regression), we can observe that the SEAL framework performs in a much more stable manner. This paper not only shows that different energy network variants can be successfully applied as a loss function, but it also shows that it is more beneficial to use energy networks as a loss function. We believe this is a nontrivial contribution.

---

> ### Author Response · Authors · 2021-11-21
> **Response (Q1-4) to comments and questions of reviewer CpcQ**
>
> Thanks for your time and valuable feedback. In the comments above, we have identified two weakness-related comments (W1-2) and four questions (Q1-4) on analysis of experiment results and experiment setup. For ease of following each topic, we divide our answers into different threads: W1, W2, and Q1-4.
>
> > Q1. Why the ranking-based methods (in table 2) are significantly better than others in the Delicious and cal500 dataset?
>
> Cal500 and Delicious have a very high diversity of 1 and 0.981 (Reference: [MLC data repo](https://www.uco.es/kdis/mllresources/#Read2010)). Diversity of 1 means that each data point holds a unique label set. It seems that the exposure to multiple samples and evaluations of their relative scores in NCEranking loss function can be helpful in a high-diversity setting.
>
>
> > Q2. Why the SEAL-based methods lead by a large margin on genbase dataset? (is it something related to the nature that this is a very small dataset?)
>
> A small dataset could be a factor. However, it seems the small label size and peculiar correlations in the label space seem to be a larger factor. Upon analysis, out of 27 labels, we found that 6 labels only occur as a singleton (by itself), 10 labels only occur as non-singleton, and 7 labels do not participate at all. This leaves only 4 labels occurring by themselves as well as with others. Not only that, only 7% (35/500) of training instances have more than one active label. In this peculiar setting, we believe feedforward networks with CE loss can easily learn singletons, and energy networks would easily learn co-occurrence, but learning both might be confusing for these two models at the opposite end of the spectrum. We believe the synergy of CE and a loss from the energy network enables capturing the best of both worlds and achieving a near-perfect score that none of the approaches by itself achieves.
>
> Another interesting fact is that CE results are surprisingly sensitive to the random seed. For CE, the F1 dropped significantly (from 75->50) after the 10-random-seed average.
>
> > Q3. Why DVN performs so well on cal500? (seems to be another small dataset).
>
> As mentioned before, Cal500 and Delicous have some similarities. They both have high diversity and also have high cardinality ( average number of active labels per instance). Among them, we think cal500’s density really stands out compared to other datasets. Density means “relative %”  of active labels per instance. While most of the datasets we studied have 1-5% density, Cal500 has a density of 15%. However, it is hard to conclude whether density is indeed the factor as we cannot find similar datasets (continuous feature, high diversity, and high cardinality) with different densities.
>
>
> > Q4. What if you replace the adapter-based BERT encoder with a full finetuning on BERT?
>
> Based on the literature on BERT-adapters  [Houlsby et al., 2019](https://arxiv.org/pdf/1902.00751.pdf)[Pfeiffer et al., 2021](https://arxiv.org/pdf/2005.00247.pdf), we find that there isn't a clear winner in terms of performance between BERT and BERT-adapter. In both of these representative adapter papers, the overall performance for BERT-adapter is very similar (-0.4 F1 and +0.64F1) to the full-finetune of BERT. To confirm whether this fact transfers to our case as well, we check the cross-entropy result for three text datasets we experimented on. Two datasets perform +0.87, +0.89 F1 points better (BGC, RCV, respectively)   and another dataset (NYT, larger data) performs -1.79 F1 points worse, which confirms our observation that there is no clear winner. Based on the results in the adapter papers and our confirmation, we see little value in increasing memory consumption and computation time for SEAL methods.
>
> Side note: (orthogonal topic) Note that SEAL-NCE with BERT-adapter performs similarly on BGC, RCV, and significantly better on NYT compared to the full BERT-base CE model.
>
> While there is no clear winner between BERT and BERT-adapter in terms of performance, there is a clear winner in terms of memory and computational efficiency. We argue that we presented an efficient way of utilizing multiple BERT models together in the SEAL framework with minimal extra memory. SEAL framework requires two input (text or feature) encoders, one for energy network ($\mathbf{F}_E(x)$) and one for feedforward network ($\mathbf{F}_F(x)$). If one were to use full BERT for both encoders, it would require more than twice the memory we are using now considering the parameters and gradient calculations. Therefore, we see the presentation of BERT-adapter in our work as a contribution that can ease the usage of pre-trained models in the proposed SEAL framework.
>
> **Reference**
> * Houlsby et al., Parameter-efficient transfer learning for NLP, 2019 ICML
> * Pfeiffer et al., AdapterFusion: Non-Destructive Task Composition for Transfer Learning, EACL 2021

---

> ### Author Response · Authors · 2021-11-21
> **Connections to the suggested related work**
>
> Thanks for your time and valuable feedback. We have identified one extra question you have made regarding SEAL's relation to "label embedding" methods.
>
> > Regarding related work, could you also discuss how "label embedding" (https://arxiv.org/abs/1503.08677) and "joint learning of label embedding and the feedforward network" (e.g., https://arxiv.org/pdf/1805.04174.pdf) relate to your proposed work here.
>
> Thank you for pointing out interesting related work. Following is our understanding of the suggested related work (with a few more of them we found in related topic):
> * [[Akata et al.,2015](https://arxiv.org/pdf/1503.08677.pdf)]: Expressing the probability of a label given input as a dot product between input and label representation.
> * [[Wang et al., 2018](https://arxiv.org/pdf/1805.04174.pdf)],  [[Xiao et al., 2019](https://arxiv.org/pdf/2106.03103.pdf) ], [[Zhang et al., 2021](https://arxiv.org/abs/2106.03103)]: These works utilize label embeddings so that the representation at final layers is a joint function of input and label representations. All of them compute the attention of a certain input or label by computing the dot product between input and label space and call this attention as label-based [Wang et al., 2018], label-specific [Xiao et al., 2019], cross-attention [Zhang et al., 2021].
>
> While the idea of utilizing label embedding seems interesting, we argue that our proposed SEAL framework and the utilization of label embedding to be an orthogonal approach for the following reason.
>  The works that utilize label representation make the input representation richer by making the final representation to be a function of input and label representation. We think this is an interesting direction, however, their final losses are simply cross entropy which treats each output probability conditionally independent given representation provided to the final layer.
>
> **What is unique in SEAL, and missing in the works mentioned above, is that the gradient value to the certain label will depend on all the labels' probability** (please refer to Appendix B that we have added). The related work above, on the other hand, utilizes the label embedding to provide richer features to the model but the loss function is the same as standard cross-entropy which does not capture relation to other label activation probabilities. Based on this fact, one could replace the feedforward network used in SEAL with the neural networks used in related work above; one could modify the energy network in SEAL by appending label embeddings on top of the probability values. As two methods can be used on top of another, we argue that SEAL is orthogonal to the aforementioned related work.
>
> **Reference**:
> * Akata et al., Label-Embedding for Image Classification, PAMI2015 [[Akata et al.,2015](https://arxiv.org/pdf/1503.08677.pdf)]
> *Wang et  al., Joint learning of label embedding and the feedforward work, ACL2018 [[Wang et al., 2018](https://arxiv.org/pdf/1805.04174.pdf)]
> *Xiao et al., Label-specific document representation for multi-label text classification, EMNLP 2019. [[Xiao et al., 2019](https://aclanthology.org/D19-1044/)]
> *Zhang et al., Enhancing Label Correlation Feedback in Multi-Label Text Classification via Multi-Task Learning, ACL 2021 Findings.[[Zhang et al., 2021](https://arxiv.org/abs/2106.03103)]

---

> ### Comment · Reviewer_CpcQ · 2021-11-29
> **Acknowledgment of author's response**
>
> Thanks for the author's response especially the answers to my questions and related work. I have updated my rating.

---

> > ### Author Response · Authors · 2021-11-29
> > **Thank you for taking your time to go through our response**
> >
> > We appreciate the reviewer CpcQ for going through our response and updating the rating accordingly.

---

### Official Review · Reviewer_vt8f · 2021-11-03

**Correctness:** 3
**Technical Novelty And Significance:** 3
**Empirical Novelty And Significance:** 3
**Recommendation:** 6
**Confidence:** 3

**Main Review:**

The general idea of SEAL is interesting since various energy-network-learning objectives can be plugged into SEAL. Such as various losses of margin-based, regression-based and contrastive-sampling-based. Figure 1 is intuitive for understanding the major updates of the network structures in SEAL. Generally, this paper is well written with good motivations, technical details and rich experiments.



Detailed questions and comments:
1.	In algorithm 1, “kaiming initialization” stands for? Do you mean Kaiming He’s method? Can you give a citation as well? What will happen if you use other types of initialization methods?
2.	In table 2, any detailed information about parameter sizes, training time and inference time comparisons among the baselines and SEAL variants? Any other types of networks that are not related to energy network to solve the tasks in table 2?  [the appended information from the authors answered this question in detail and thanks for that.]


**Summary Of The Paper:**

This paper proposes a framework named SEAL that can adopt structed energy network as both a trainable and adaptable loss function for training feedforward networks. 7 feature-based 3 text-based datasets are used for testing SEAL and showed better results and even better with combinations. Rich ablation study was given as well.

**Summary Of The Review:**

Strong:
1.	The idea of SEAL is novel and effective to a number of tasks;
2.	This paper is well written and the direction of using of dynamic loss functions is an interesting direction.

Weak:
1.	Less aware of the signicient technical aspects of using energy network for the target tasks. Any other baselines of not using energy related networks in table 2? [thanks the authors' responses for the detailed explanation of this point. Now I have a better understanding of the benefits of applying "energy-based methods" to the corresponding tasks.]

---

> ### Author Response · Authors · 2021-11-20
> **Response to Q2 for detailed information about parameter size, training and inference time.**
>
> > Q2.  Parameter size and training/inference time comparisons.
>
> Thank you for raising this interesting question. Here, we present a number of parameters involved in training time (first table) as well as train and inference time consumption comparisons.
> In summary, SEAL requires approximately twice as large parameters compared to other losses and is slower in training time. However, in inference time, SEAL is as fast as the CE method and requires an equal number of parameters involved. We will include these observations and the tables below in the appendix of the paper.
>
> **Number of parameters**
>
> The number of parameters required during train time for SEAL is approximately double the size of feedforward (CE column) while energy network and feedforward sizes are comparable. However, in the inference time, SEAL has an equal amount of parameters to the CE column as the energy network in SEAL is only utilized during training.
>
> | Parameter size | CE      | Energy network (SPEN, DVN, NCE) | SEAL (train) | SEAL (inference) |
> |----------------|---------|---------------------------------|--------------|------------------|
> | EXPR_FUN       |  333000 |                          533800 |       866800 |           333000 |
> | SPO_FUN        |  447000 |                          597600 |      1044600 |           447000 |
> | Bibtex         |  958800 |                          991000 |      1949800 |           958800 |
> | Cal500         | 1158700 |                         1123500 |      2282200 |          1158700 |
> | Delicious      |  754000 |                          951000 |      1705000 |           754000 |
> | Genbase        |  485600 |                          491400 |       977000 |           485600 |
> | Eurlex-ev      | 4747500 |                         5546500 |     10294000 |          4747500 |
>
> **Inference time (per epoch)**
>
> We simply average inference time for CE and SEAL variants as they are very similar. Likewise, we average the inference time of different energy networks. Here, inference time (sec) is recorded for the whole validation set. We also present speed per sec (example/sec) and speed ratio. The last column shows that CE and SEAL methods are 2x-7x faster in inference time than the energy networks.
>
> | Inference time/speed | Energy network (sec) | CE and SEAL (sec) | Energy network (example/sec) | CE and SEAL (example/sec) | Speed ratio (CE/EnergyNetwork) |
> |----------------------|----------------------|-------------------|------------------------------|---------------------------|--------------------------------|
> | EXPR_FUN             |                 1.33 |              0.22 |                          638 |                      3801 |                           5.96 |
> | SPO_FUN              |                 1.18 |              0.16 |                          709 |                      5231 |                           7.38 |
> | Bibtex               |                  3.6 |              1.78 |                          414 |                       840 |                           2.03 |
> | Cal500               |                 0.24 |              0.10 |                          438 |                      1080 |                           2.47 |
> | Delicious            |                 5.35 |              1.39 |                          599 |                      2307 |                           3.85 |
> | Genbase              |                 0.23 |              0.13 |                          574 |                      1005 |                           1.75 |
> | Eurlex-ev            |                   24 |             12.22 |                          162 |                       317 |                           1.96 |
>
> ** Train time (sec/epoch) **
> The training time per epoch is presented per dataset and per loss function used. Due to different gpu types and node status, there are some outliers. However, the general trend is: CE < Energy Networks < SEAL.
>
> | Time (sec per epoch) | EXPR_FUN | SPO_FUN | Bibtex | Cal500 | Delicious | Genbase | Eurlex-ev |
> |----------------------|----------|---------|--------|--------|-----------|---------|-----------|
> | CE                   |     6.29 |    5.46 |  22.67 |   1.00 |     33.01 |    2.12 |    114.84 |
> | SPEN                 |    10.99 |   11.01 |  28.04 |   2.53 |     37.31 |    3.57 |    136.10 |
> | DVN                  |    10.95 |   10.02 |  32.12 |   1.87 |     55.24 |    2.79 |    129.94 |
> | NCE                  |     3.71 |    3.83 |  13.97 |   2.82 |     22.03 |    3.68 |     89.33 |
> | margin               |    27.96 |   35.24 | 212.94 |   5.78 |     44.60 |    8.73 |    260.10 |
> | regression           |    46.32 |   56.97 |  27.78 |   8.07 |    143.90 |   13.91 |    352.63 |
> | regression-s         |    42.87 |   77.92 | 179.97 |   8.48 |     45.33 |   10.79 |    221.33 |
> | NCE                  |    72.43 |   40.34 | 131.73 |  16.58 |    158.97 |   17.77 |    431.92 |
> | Ranking              |    41.24 |   26.83 | 218.37 |   7.04 |    317.63 |   10.74 |    408.79 |

---

> ### Author Response · Authors · 2021-11-21
> **Response to Q1 and W1.**
>
> Thank you for your time and feedback. We identified two questions (Q1 initialization, Q2 runtime, parameter size) and one weakness (W1) from the reviewer. As answers of Q2 requires large tables, we separate the response into two parts: Q1, W1 in one part and Q2 in another part.
>
> > Q1. Algorithm 1. Initialization method.
>
> Thanks for pointing out the missing reference. To answer your question, yes, it is  He, K et al. (2015) as [pytorch documents](https://pytorch.org/docs/stable/nn.init.html) shows. However, we will remove this mention of ‘Kaiming initialization’ in the Algorithm1 as we realize it could mislead readers into thinking that our method depends on this specific initialization. This is just one of the experiment setups we tried and
>  SEAL framework can work with any initialization method. As Kaiming initialization is one of the standard approaches for initialization,  we don't plan on studying the effect of various parameter initializations in this work as we consider it is a parallel research direction.
>
> > W1 Less aware of the significant technical aspects of using energy network for the target tasks. Any other baselines of not using energy related networks in table 2?
>
> **Clarification**:
> To properly answer this question, we first ask the reviewer whether this question is intended to ask “why this paper does not compare with other multi-label classification models that are non-energy-based approaches”. We present our response assuming that our understanding of the question is correct, please let us know if we misinterpreted your question.
>
> **Response**:
> First, the energy-based approaches (SPEN, DVN) that we compare to in our paper have emerged as the most successful methods for the feature-based multi-label classification task. Since we further improve the best performing energy methods (SPEN, DVN), we simply compare with them, rather than comparing with previous baselines of SPEN and DVN: full pairwise potential methods [[Chen et al., 2015](http://proceedings.mlr.press/v37/chenb15.pdf )], [[Schwing et al., 2015](https://arxiv.org/pdf/1503.02351.pdf)] and posterior regularization  [[Lin et al., 2014](http://victorialin.net/pubs/prlr_mlnlp_2014.pdf) ].
>
> Second, the main focus of this paper is in comparing different dynamic loss functions that emerge through the use of energy-network. In particular, SEAL asks whether a loss function that jointly scores the whole high-dimensional output probability vector (via energy network) is beneficial compared to taking each label conditionally independent as CE loss does. In that perspective, most of the work we found in text-based MLC studies the orthogonal topic of leveraging label embeddings as an extra feature to the model [[Wang et al., 2018](https://arxiv.org/pdf/1805.04174.pdf)],  [[Xiao et al., 2019] (https://arxiv.org/pdf/2106.03103.pdf)], [[Zhang et al., 2021](https://arxiv.org/abs/2106.03103)]. While aforementioned works try to create richer representation via incorporating label embeddings, the loss they resort to in the end is the standard CE loss. As the SEAL approach can be applied to the architecture of these related works, and as injection of label embedding can be applied to energy networks, we view the techniques applied in related works as complementary to the SEAL framework. Hence, we do not consider these as our baselines.
> (For a more detailed description of the related work in MLC, please refer to our response on Q5 of reviewer CpcQ)
>
> Lastly, our text MLC  performances are competitive and comparable to other papers. The capsule network on BGC performs 74.3 F1 [[Aly et al., 2019](https://aclanthology.org/P19-2045)](Table 2) whereas the BERT-adapter model with CE and SEAL-NCE has 81.15 F1 and 81.64 F1,  respectively.  [[Zhang et al., 2021](https://arxiv.org/abs/2106.03103)](Table 2) improves RCV F1 from 87.7 to 88.5, whereas our CE vs. SEAL-NCE gets 87.18 to 87.82. On the RCV dataset, due to train size* mismatch, there is a slight performance difference, but we argue that this is a competitive performance range to study the effect of the loss function.
>
> *On the RCV dataset, the slight performance difference might be from the fact that we are using official training set of RCV1-V2 which is a significantly smaller training set (1.7%) compared to that of [Zhang et al., 2021].
>
> **References**
> * Lin et al., Multi-label learning with posterior regularization, 2014 NeurIPS workshop
> *Chen et al., Learning Deep Structured Models, 2015 ICML.
> *Schwing et al., Fully connected deep structured networks. arXiv 2015
> *Aly et al., Hierarchical Multi-label Classification of Text with Capsule Networks, ACL 2019
> *Wang et al., Joint learning of label embedding and the feedforward network, ACL 2018
> *Xiao et al., Label-specific document representation for multi-label text classification, EMNLP 2019
> *Zhang et al., Enhancing Label Correlation Feedback in Multi-Label Text Classification via Multi-Task Learning, ACL 2021 Findings.

---

> ### Author Response · Authors · 2021-11-30
> **Looking forward to final comments**
>
> Since the discussion period closes soon, we wanted to ask the reviewer if they have any comments based on our previous responses that include additional information like number of parameters and training/inference times.

---

> > ### Comment · Reviewer_vt8f · 2021-11-30
> > **thank you so much for the detailed information**
> >
> > Thanks a lot for the detailed training time, parameter size and inference time information.
> > Please allow me to take a moment to update my former review messages accordingly.

---

### Official Review · Reviewer_5BUz · 2021-11-04

**Correctness:** 3
**Technical Novelty And Significance:** 3
**Empirical Novelty And Significance:** 3
**Recommendation:** 6
**Confidence:** 4

**Main Review:**

Strengths:
- Interesting approach for solving structured prediction problems that shows significant improvement in performance (F1 score) on feature-based datasets
- A thorough experimental analysis, including important ablation study, that demonstrates the strengths of the proposed approach.


Weaknesses:
- The proposed approach is a generalization of [Tu et al., 2020] (that is represented by the margin-based SEAL version). The proposed approach is not restricted to margin-based loss and indeed the new loss functions outperform margin-based loss, however the performance gain is smaller compared to the performance gain over non-SEAL approaches.
- I have some concerns about the reporting of the experimental results:
	1) The description of the experimental results is a bit misleading due to the focus on the performance gain over cross entropy (e.g., "+50 F1") in  Section 4 and the abstract. CE is not the state-of-the-art, and the performance gain over the state of the art should be reported (which, as far as I understand from the work, is margin-based SEAL that appears in [Tu et al., 2020]).
	2) Results on large text datasets only contain SEAL with NCEranking loss. It is not clear why the other losses (including the one that represents [Tu et al., 2020]) and other existing approaches like SPEN and DVN are not reported. It is hard to evaluate the gain of SEAL over these approaches.
- The paper only considers one type of structured prediction benchmarks: multi-label classification. It would be interesting to show the performance on additional tasks like POS tagging and NER.
- Some related literature that is missing: recent works that have used NCE for training energy based models for text-based datasets [1, 2].


Minor comments, typos, etc:
- page 1 abstract line 1 "this"
- page 1 abstract "compared" -> "Compared"
- page 2 "network  using": looks like double space
- page 2 "as efficient at inference as well as a feed-forward ....": "as well as" -> "as"
- page 2 "Despite these efforts, .... remain relatively inefficient" and "are finicky to train" - need more details or relevant citations
- page 4 "(Ma & Collins)" - missing year
- page 9 conclusion: "BLEURTSellam"


[1] Bhattacharyya, S., Rooshenas, A., Naskar, S., Sun, S., Iyyer, M., & McCallum, A. (2020). Energy-based reranking: Improving neural machine translation using energy-based models. arXiv preprint arXiv:2009.13267.

[2] Bakhtin, A., Deng, Y., Gross, S., Ott, M., Marc'Aurelio Ranzato, & Szlam, A. (2021). Residual Energy-Based Models for Text. J. Mach. Learn. Res., 22, 40-1.


**Summary Of The Paper:**

The paper presents Structured Energy As Loss (SEAL) that uses structured energy networks as a trainable loss for a feedforward network. SEAL is a general framework that supports multiple loss functions, including margin-based loss, regression-based loss, and NCE-based loss, and generalizes the recent approach in [Tu et al., 2020]. The paper presents extensive experiments on multi-label classification on a range of feature-based and text-based datasets. SEAL shows significant improvement over standard cross entropy trained networks, as well as approaches based on Structured Prediction Energy Networks (SPEN).

**Summary Of The Review:**

Overall, I think the paper presents an interesting approach that shows significant improvement in multi-label classification. However, the experimental analysis could be strengthen by improving the presentation of the results, and by considering more benchmarks.

---

> ### Author Response · Authors · 2021-11-12
> **Thanks for your time and thoughtful comments. One question on point W2.**
>
> Thanks for your time and thoughtful comments. We plan to write a more thorough response addressing all your comments, but before that, we wanted to first follow up on your comments that might require more experiments. Particularly, we would appreciate reviewer 5BUz’s opinion on whether extra experiments on SPEN, DVN on a larger text dataset would be beneficial for our paper. We post that discussion on W2.
>
> > W1 “Performance gain over margin-based SEAL is smaller”.
>
> Yes, as mentioned in the abstract, SEAL-NCE gets +2.85 F1 gain compared to CE and +2.23 F1 gain compared to margin-based SEAL (which is our run of Tu et al. 2020). This extra gain of +2.23 over SEAL-margin is far larger than SEAL-margin’s gain over CE which is +0.62 F1 point.  To make the comparison easier, we provide a table that compares the average F1 point difference between SEAL-NCE and other SEAL variants here.
>
> |   Average F1 differences  | With Genbase | Excluding Genbase |
> |---------------------------|--------------|-----------------|
> | SEAL-NCE - SEAL-margin | 2.02       | 2.23            |
> | SEAL-NCE - SEAL-regression       | 1.58        | 1.78            |
>
> We argue that the gain of +2.33, on average, is pretty significant in the given context. Do you feel that the gain of +2.23, on average ( excluding Genbase outlier) is too small?
>
> > W2 "Why run SEAL-NCE only on large text dataset?"
>
> First of all, apologies for not making it very clear as to why we did not run other variants of SEAL and other versions of the energy network. The three main reasons for this choice are:
>
> **Reason 1**: The results shown in table 2 provide a good amount of evidence about the general superiority of SEAL-NCE over other SEAL variants. Hence, considering the computational expense of running hyper-parameter searches on BERT for various loss functions we consider, we chose to compare SEAL-NCE with cross-entropy baseline, as it is the most common baseline for text classification.
>
> **Reason 2**: Energy networks, i.e. SPEN and DVN, on their own, did not perform very well-- SPEN, DVN only outperformed CE on 2, 4 datasets, respectively, out of the 7 datasets. As seen in the Table 2, the DVN and SPEN results have a large variance in their comparative performance to CE models and also they were performing much worse than SEAL models, we thought they would not add much value to run on larger text datasets as well. The table at the end summarizes the points we make.
>
> **Reason 3**: Albeit being evaluated on a different task, Table 7 of [Tu et al. 2020b](https://aclanthology.org/2020.emnlp-main.449.pdf) observed only a marginal improvement (+0.01, +0.33 F1 point) over the BERT model trained using CE. With this result and the performance gap between  SEAL-NCE and SEAL-margin on the feature-based datasets, we thought it is not necessary to run SEAL-margin again with BERT-adapter as we thought the two had a clear trend.
>
> Tu L, Liu T, Gimpel K. An Exploration of Arbitrary-Order Sequence Labeling via Energy-Based Inference Networks. EMNLP 2020b
>
> Due to the reasons mentioned above, we decided to not run [Tu et al. 2020], SPEN, and DVN. However,  upon reading your review, we are open to reconsidering this choice. With the additional context provided here, do you still think that adding SPEN and DVN for text datasets would add significant value to the empirical analysis?
>
> |                      | Average F1 differences excluding Genbase | Number of datasets performing better than CE |
> |----------------------|------------------------------------------|----------------------------------------------|
> | SPEN-CE              | 0.12                                     | 2                                            |
> | DVN-CE               | -0.37                                    | 4                                            |
> | NCE-CE               | -15.59                                   | 1                                            |
> | SEAL-margin - CE     | 0.62                                     | 6                                            |
> | SEAL-regression - CE | 1.1                                      | 6                                            |
> | SEAL-NCEranking - CE | 2.85                                     | 7                                            |

---

> > ### Comment · Reviewer_5BUz · 2021-11-18
> > **Thank you for the response**
> >
> > Thank you for the response. Regarding your question on W2:
> > I understand the points you made and I think it is important to revise the paper to reflect the reasons you mentioned. I still think that it would be useful to see the results for [Tu et al., 2020] as the results on a different task may not be indicative of the performance on this task.

---

> > > ### Author Response · Authors · 2021-11-19
> > > **SEAL-margin results**
> > >
> > > Thank you for answering our question. As the reviewer suggested, we will revise the paper to clearly explain our reasonings stated above.
> > >
> > > Upon leaving the previous comments, we were convinced that there is value in running SEAL-margin (or [Tu et al. 2020]) and here we report the result in the below table. [Tu et al., 2020] performs on par or better than CE, and SEAL-NCE have about 0.5-0.8 F1 point improvement over [Tu et al., 2020] across three datasets. The scale of improvement reduced a bit, nonetheless, the overall trend of experiments in text datasets on BERT adapter seems to be similar to experiment results feature-based datasets with MLP layers: SEAL-NCE > SEAL-margin > CE.
> > >
> > > | Method \ Data Name                | BGC   | RCV   | NYT   |
> > > |-----------------|-------|-------|-------|
> > > | CE              | 81.15 | 87.18 | 77.40 |
> > > | SEAL-margin     | 81.14 | 87.01 | 78.13 |
> > > | SEAL-NCEranking | 81.64 | 87.82 | 78.87 |
> > > (All values in the table are $F_1$ scores.)
> > >
> > > Side note: After the SEAL-margin experiment, we tried to conduct experiments with DVN as well, one of the better-performing energy networks, on these text datasets. However, due to GPU server failure, we were not able to get the exact numbers in time. If we get the result in time (before Nov 22nd), we will also include the DVN result in the paper.

---

> ### Author Response · Authors · 2021-11-20
> **Response to remaining topics.**
>
> Thank you for your time and effort. We have previously discussed two weakness points (W1-2) and presented the requested experimental result. We now delve into the remaining topics (W3-5, S1).
>
> >W3 “On reporting of experimental results and comparing to CE”
>
> Our goal was to compare a loss function that treats output probabilities conditionally independent (CE) and SEAL losses which take all the label probability vectors together.
> Given the result presented in Table 2 along with the description “*average gains over CE ranging from +0.62 (SEAL−margin) - +2.85 (SEAL-NCEranking) F1 points in average when the CE results are around 30-50 F1*”, we thought the fact that average gain of SEAL-NCEranking over SEAL-margin, which is 2.23 (2.85-0.62), would be obvious.
>
> While we think the gains between each SEAL variant are clear in table 2, I also can see concerns that you might have: proper comparison with the previous method. **Do you think adding the following sentence could clarify things a bit more? (We plan to add this sentence in the updated pdf)**
> “The best SEAL variant, SEAL-NCE gains +2.23 (without genbase) and  +2.03 (with genbase)  over the previous approach of Tu et al. 2020, which is in SEAL-margin in table 2.”
>
> > W4 “Mention of +50 F1 on genbase.”
>
> We exclude genbase from our average gain calculation because it is an outlier. For this, we write in the  abstract, “*+2.85, +2.23 respective F1 point gain in average over cross-entropy and INFNET the feature-based datasets, excluding one outlier that has an excessive gain of +50.0 F1 points.*”. Here, the purpose of mentioning the gain of +50 F1 points on genbase is not to mislead the reader, but provide them with complete information, i.e., we consider genbase dataset to be an outlier because of the model’s unusually high performance on it. **That being said, do you think this point will be better made if we do not mention the exact gain on the genbase dataset but only state that it is an outlier?**
>
> Also, based on your comment and reviewer CpcQ’s comment, along with pointing out the fact that genbase dataset behaves as an outlier, we now also attempt to explain the reason behind the unusually high performance of our model on it. Please refer to our response to CpcQ for the explanation.
>
> > W5 Running on different tasks as well?
>
> We are definitely interested in extending our work to other tasks such as sequence tagging problems like POS, NER as you suggested. As you have noted, this structure is very general and can be used for sentence-level classification as shown in this work, as well as other structure prediction tasks like sequence tagging, SRL, semantic parsing etc.
>
> Nonetheless, since the SEAL framework has a lot of design choices, before trying the framework on more complex tasks, we first wanted to perform a thorough analysis of the effect of various possible loss functions under the SEAL framework. Doing this through a single task on multiple datasets, seemed like a better choice and hence MLC was chosen as the sole task for this paper as SPEN, DVN and first INFNET [[Tu et al.2019](https://arxiv.org/pdf/1803.03376)] commonly studied this problem.
>
> The design choices of SEAL include selecting one hyper parameter from various hyperparameters such as respective number of layers, hidden dimension for the feature network of feedforward and energy network. By focusing on MLC only, we have conducted very rigorous hyperparameter search for each configuration, running 200+ hyperparameter search for each setup and then reporting average performance over 10 runs utilizing different random seeds. We carefully designed the experiments to have enough large labels as other MLC papers have [[Aly et al., 2019](https://aclanthology.org/P19-2045)], [[Xiao et al., 2019](https://aclanthology.org/D19-1044/)], [[Zhang et al., 2021](https://arxiv.org/abs/2106.03103)]. Additionally, we did not forget to have diversity on data size (small, medium, large) and data types  (continuous-feature, discrete-feature, and variable length text data) to examine SEAL in different angles.
>
> On top of the choices mentioned above, using SEAL for complex structured prediction tasks would add other design variables like choice of task specific network architectures for the energy, sampling strategy (specifically for SEAL-NCE), etc. As these are orthogonal to the focus of the current paper, we leave it for future work.
>
> > S1. Suggestion of related work
>
> Yes, we will include the suggested related work as an application of energy models in NLP applications that use energy scores directly to rerank as opposed to our method that utilizes gradient information.
>
> > S2. Typos, etc.
>
> Thank you for catching these errors. We have reflected all your suggestion in our document, including the suggestion to add a citation to the sentence in the introduction (page 2."Despite these efforts ... remain relatively inefficient").

---

> > ### Comment · Reviewer_5BUz · 2021-11-30
> > **Thank you for your response**
> >
> > I thank the authors for their thoughtful response. The response addresses my concerns regarding the results on text datasets. I think that the proposed approach shows significant gains in performance (especially on the feature-based datasets).
> >
> > My main concerns remain the limited novelty (w.r.t recent approach [Tu et al., 2020]) and the experiments on one task - multi-label classification (unlike, for example, [Tu et al., 2020] or DVN that include experimental results for more than one task). I therefore keep my score of 6.

---

> ### Author Response · Authors · 2021-11-30
> **Looking forward to the final comments**
>
> Since the discussion window closes soon, we wanted to ask the reviewer if they have any final comments based on our previous responses and provided results.

---

### Author Response · Authors · 2021-11-25
**General response**

We sincerely thank the reviewers for their time, effort, and helpful feedback. In this general response, we first summarize the changes made to the pdf based on the reviews, and then we highlight the contributions of this work in terms of novelty and impact

We have made the following changes in the final rebuttal revision:
- Fixed minor typos and grammar errors. (Including ones in the reviewer feedback)
- Added how the gradient from an energy network differs from that of cross-entropy in Appendix B.
- Added the inference time, parameter sizes, and train time analysis in Appendix E. (vt8f’s request)
- Added related works (per 5BUz and CpcQ's request).
- Added extra experiment result for SEAL-margin on the text-based dataset and clarified on why we do not run experiments for all setups in the text-based dataset. (5BUz's request)

Now, we outline three major points that we want to deliver in the general response.

**1. Novelty: shift of perspective**

The first contribution of our paper is the change of perspective on how one can utilize an energy network.
To the best of our knowledge, this is the first work that presents a general concept of teaching feedforward networks (such as MLP or BERT) through minimizing energy defined by learned energy networks. Our main question was on whether an energy network can capture rich relationships in the output space that cross-entropy cannot capture, and function as a good evaluator of model output to further train it. Our contribution is proposing this shift of perspective as well as showing the generality of SEAL across different energy network variants.

We also differentiate SEAL from the previous approach [Tu et al., 2018,  Tu et al., 2020] in section 3 of the paper and in our response to 8Y6b ([W2](https://openreview.net/forum?id=dEOeQgQTyvt&noteId=0TwZvH2TaPj)).

**2. Novelty of NCE approach we proposed.**

We propose a unique NCEranking loss for SEAL motivated by the loss in [Ma & Collins, 2018]. Our contributions are as follows. First, we utilize NCE to provide a gradient toward the feedforward model within the SEAL framework rather than using it as a scoring function for reranking outputs [Bhattacharyya et al., 2021, Deng et al., 2020]. Second, we provide a symbiotic structure where the NCE energy model improves together with the noise model (our feedforward network). Whereas conventional approaches utilize a fixed noise model, in the SEAL framework, the NCE model teaches a noise model and the noise model provides samples for the NCE model in an alternating fashion.
To the best of our knowledge, both contributions have not been addressed prior to our paper.

To show the effectiveness of this approach, we show that **NCE loss is not very helpful on its own without the SEAL framework, neither for an energy network nor for a feedforward network.** As shown in the *energy only* part of Table 2, the isolated energy network trained with the NCE loss performs very poorly. As shown in the ablation study (Sec 5. *Effect of applying ranking loss directly on $F_\phi$*), the ranking loss is also not very useful when it is directly applied to the feedforward network. In our experiments, the proposed SEAL-NCE was the only setup that gave a high boost in performance and we argue that this is a novel contribution.

**3. Need to evaluate the model with larger datasets.**

> **CpcQ**:  The impact of this paper is limited because the experimental dataset are mostly on a label cardinality under 1000 ...  author could use a larger dataset and a larger label cardinality.

>**8Y6b**: compare utilizing existing datasets from past works?

To answer these points, we reused the response to reviewer 8Y6b ([W4](https://openreview.net/forum?id=dEOeQgQTyvt&noteId=yENSBD1DDgR)) for other reviewers.

 **SEAL is tested on three datasets with label cardinality equal greater than 1000: Delicious with 1k, NYT with 2k, and Eurlex with 4k**.  This is a larger number of high-cardinality data sets compared to previous works (For more details, refer to our response to CpcQ ([W1](https://openreview.net/forum?id=dEOeQgQTyvt&noteId=5QDloza0Ry))).  Moreover, we believe evaluating on small datasets is as important as large labeled data is not always accessible in the real world, often referred to as a low-resource problem. With these facts in mind, we argue that our portfolio of experiments is more comprehensive than other MLC or energy network papers.

We also **utilize existing datasets from past works**:  Bibtex and delicious, which were utilized in the past work, have been included in our experiments for making comparisons with existing models like SPEN, DVN, and INFNET. Lastly, we examine two datasets previously utilized in the text MLC problem:  BGC and RCV. We argue that the level of performance is competitive and comparable to other papers. (For more details, refer to our response to 8Y6b (W4))

---

> ### Author Response · Authors · 2021-11-25
> **References on above post**
>
> We provide references used in the general response above.
>
> * [Lifu Tu, Kevin Gimpel, Learning Approximate Inference Networks for Structured Prediction, ICLR 2018](https://arxiv.org/abs/1803.03376)
> * [Lifu Tu, Richard Yuanzhe Pang, Kevin Gimpel, Improving joint training of inference networks and structured prediction energy networks. In SPNLP, 2020.](https://arxiv.org/abs/1911.02891)
> * [Deng Y, Bakhtin A, Ott M, Szlam A, Ranzato MA. Residual energy-based models for text generation, ICLR 2020](https://openreview.net/pdf?id=B1l4SgHKDH)
> * [Bhattacharyya S, Rooshenas A, Naskar S, Sun S, Iyyer M, McCallum A. Energy-based reranking: Improving neural machine translation using energy-based models. ACL 2021](https://aclanthology.org/2021.acl-long.349.pdf)

---

### Decision · Program_Chairs · 2022-01-20

**Decision:**

Reject

**Comment:**

This paper proposes a new method for multi-label classification, which leverages the advantage of the emery-based model. However, one reviewer and the area chair have two serious concerns on the experiments: (1) The proposed method is only evaluated on low dimensional datasets; (2) Some important baselines methods are missing, which makes the comparison not convincing. I suggest the authors to evaluate their methods on more datasets, and add the results from well known multi-label classification method for comparison.

---

> ### Public Comment · ~Jay-Yoon_Lee1 · 2022-02-15
> **Reply to AC from authors**
>
> We value the AC’s input; however, we believe that we have already addressed all the concerns of AC and feel that our responses have been ignored. As noted in our general response ([item3](https://openreview.net/forum?id=dEOeQgQTyvt&noteId=tI65NW1nn_A)) this paper examines more datasets (3 text datasets and 7 feature-based datasets) than usual text multi-label classification (MLC) papers (2-5 datasets in our survey). We also do not use lower-dimensional data both in label and input dimensions compared to other MLC papers. For more details, we again summarize our previous responses in the below comment. Reviewer CpcQ first brought up this topic of small data and small labels; however, they raised their rating after considering our response ([link-w1](https://openreview.net/forum?id=dEOeQgQTyvt&noteId=QRjjKFmEFgW)). As mentioned in response to reviewers ([link-w1](https://openreview.net/forum?id=dEOeQgQTyvt&noteId=QRjjKFmEFgW), [link2-w4](https://openreview.net/forum?id=dEOeQgQTyvt&noteId=yENSBD1DDgR)), this paper demonstrates competitive performance compared to other papers that have a specific architecture for the MLC problem. As we asked reviewer 8Y6b, we ask the AC why they consider these references not to be a good comparison.
>
> ---
>
> **More details from previous response and questions to AC**
>
> AC had three major comments: (1) the author should test on more datasets, (2) the datasets currently used are “low-dimensional”, and (3) other “well-known” MLC baselines are missing for comparison. Despite the use of somewhat vague terms like “low-dimensional” and “well-known” in the review, we have, to the best of our ability, tried to resolve these concerns in our responses. Since the AC has used the same terms despite our responses addressing them, we would appreciate it if the AC could elaborate more on the following aspects (our previous responses are summarized alongside).
>
> **1. What more datasets should we run on?**
> We evaluate our models on more datasets than average MLC papers. The papers listed in response ([[Aly et al., 2019](https://aclanthology.org/P19-2045)], [[Xiao et al., 2019](https://aclanthology.org/D19-1044/)], [[Zhang et al., 2021](https://arxiv.org/abs/2106.03103)] ) usually examine 2-5 text-based datasets. In our experiments, we examine our method on 3 text-based datasets as well as add 7 more feature-based datasets (considering different label sizes, domains, and feature types).  We ran feature-based datasets used in other energy network works and text-based datasets from other text MLC papers as explained in [link2-w4](https://openreview.net/forum?id=dEOeQgQTyvt&noteId=yENSBD1DDgR). On top of existing data from other papers, we expanded the feature dataset to be more diverse for extensive experiments.
>
> **2. What do they mean by low dimensional data?**
> We are not sure what AC meant by “low-dimensional data”. If the AC meant, a low-dimensional input feature,  we examine our model with the standard text dataset (thus same input dimension as other papers) as well as with features ranging from small (70) to large (5000) dimensions. If the AC meant low-dimensional datasets as small numbers of labels, we claim that we have larger number labels than general text MLC papers. Among the four papers we explored, all of them only consider label size (<300) except one paper. For this topic, we convinced reviewer CpcQ with our response ([link](https://openreview.net/forum?id=dEOeQgQTyvt&noteId=5QDloza0Ry)).
>
> **3. What other well-known multi-label classification methods do they think we should have for the feature-baed and for the text-based?**
>
> We discussed the performance of the recent papers in our response ([link-w1](https://openreview.net/forum?id=dEOeQgQTyvt&noteId=QRjjKFmEFgW), last paragraph). Although this paper is not aimed at setting the benchmark in multi-label classification (MLC), we discuss that this paper is displaying competitive performance with other MLC-specific models. Furthermore, if one sees [[Zhang et al., 2021](https://arxiv.org/abs/2106.03103)], a simple BERT MLC model outperforms most of the previous methods – this makes our improvements on BERT MLC to be relevant to other recent MLC models.